# Phosphorylation of Arabidopsis UVR8 photoreceptor modulates protein interactions and responses to UV-B radiation

Wei Liu [1], Giovanni Giuriani[1,7], Anezka Havlikova[1,7], Dezhi Li[1,7], Douglas J. Lamont[2], Susanne Neugart[3], Christos N. Velanis [1,5], Jan Petersen[1,6], Ute Hoecker [4], John M. Christie [1] & Gareth I. Jenkins [1] ✉

Exposure of plants to ultraviolet-B (UV-B) radiation initiates transcriptional responses that modify metabolism, physiology and development to enhance viability in sunlight. Many of these regulatory responses to UV-B radiation are mediated by the photoreceptor UV RESISTANCE LOCUS 8 (UVR8). Following photoreception, UVR8 interacts directly with multiple proteins to regulate gene expression, but the mechanisms that control differential protein binding to initiate distinct responses are unknown. Here we show that UVR8 is phosphorylated at several sites and that UV-B stimulates phosphorylation at Serine 402. Site-directed mutagenesis to mimic Serine 402 phosphorylation promotes binding of UVR8 to REPRESSOR OF UV-B PHOTOMORPHOGENESIS (RUP) proteins, which negatively regulate UVR8 action. Complementation of the *uvr8* mutant with phosphonull or phosphomimetic variants suggests that phosphorylation of Serine 402 modifies UVR8 activity and promotes flavonoid biosynthesis, a key UV-B-stimulated response that enhances plant protection and crop nutritional quality. This research provides a basis to understand how UVR8 interacts differentially with effector proteins to regulate plant responses to UV-B radiation.

Ultraviolet (UV) radiation in sunlight has an extensive impact on organisms and ecosystems[1]. Short wavelength UV-B radiation has the potential to damage macromolecules and impair cellular processes and can have negative effects on many organisms, including humans. It is therefore remarkable that plants rarely show signs of UV-damage, despite continual exposure to sunlight. This is because plants can use UV-B radiation as a regulatory stimulus, enabling them to acclimate to ambient conditions to ensure UV-protection and optimal growth[2–6]. Exposure of plants to UV-B wavelengths initiates extensive changes in

gene expression that underpin a range of photomorphogenic and developmental responses, together with acclimatory modifications to metabolism and physiology, in diverse species[2–6]. Many of these responses to UV-B radiation are mediated by the photoreceptor protein UV RESISTANCE LOCUS 8 (UVR8), which senses UV-B and short wavelength UV-A light[2–7].

UVR8 exists as a homodimer in the absence of UV-B radiation. UV-B absorption by tryptophan amino acids in the primary sequence causes dissociation of UVR8 homodimers[8–11], resulting in the formation

[1]School of Molecular Biosciences, College of Medical, Veterinary and Life Sciences, Bower Building, University of Glasgow, Glasgow G12 8QQ, UK. [2]FingerPrints Proteomics Facility, School of Life Sciences, Discovery Centre, University of Dundee, Dow Street, Dundee DD1 5EH, UK. [3]Department of Crop Sciences, Division Quality and Sensory of Plant Products, Georg-August-Universität Göttingen, D-37075 Göttingen, Germany. [4]Botanical Institute and Cluster of Excellence on Plant Sciences (CEPLAS), Biocenter, University of Köln, 50923 Köln, Germany. [5]Present address: School of Life, Health and Chemical Sciences, Faculty of Science, Technology, Engineering and Maths, Venables Building, The Open University, Walton Hall Campus, Milton Keynes MK7 6AA, UK. [6]Present address: Matthias Schleiden Institute of Genetics, Bioinformatics and Molecular Botany, Friedrich Schiller University, 07743 Jena, Germany. [7]These authors contributed equally: Giovanni Giuriani, Anezka Havlikova, Dezhi Li. ✉e-mail: gareth.jenkins@glasgow.ac.uk

of monomeric UVR8, which initiates signal transduction. A small fraction of UVR8 rapidly accumulates in the nucleus following UV-B exposure[12–14], where it is able to regulate gene expression through physical interactions with other proteins. These interactions involve, at least in part, a 27-amino acid region in the C-terminus of the photoreceptor (termed C27)[15–23]. Interaction of monomeric UVR8 with the E3 ubiquitin-ligase component CONSTITUTIVELY PHOTO-MORPHOGENIC 1 (COP1) bound to a SUPPRESSOR OF PHYA-105 (SPA) protein, impairs proteolytic degradation of target protein substrates. In particular, binding of UVR8 to COP1-SPA stabilises the ELONGATED HYPOCOTYL 5 (HY5) transcription factor[24,25], which regulates tran-scription of many genes that underpin UVR8-mediated responses[24,26]. Conversely, binding of COP1 to UVR8 de-stabilises PHYTOCHROME INTERACTING FACTOR 5 (PIF5), contributing to the suppression of extension growth in UV-B[27]. UVR8 additionally binds the REPRESSOR OF UV-B PHOTOMORPHOGENESIS 1 (RUP1) and RUP2 proteins, which promote re-dimerisation of UVR8 monomers[28] and therefore act as negative regulators of UVR8 action[29]. Moreover, RUP proteins form a second E3 ubiquitin-ligase complex that degrades HY5[30]. UVR8 also interacts with several transcription factors, which modifies their ability to bind to target gene promoters, leading to positive regulation of responses to UV-B. WRKY DNA-BINDING PROTEIN 36 (WRKY36) represses transcription of the *HY5* gene, and binding of WRKY36 to UVR8 in the nucleus relieves this repression[17]. In addition, interaction of UVR8 with BES1-INTERACTING MYC-LIKE 1 (BIM1) and depho-sphorylated BRI1-EMS-SUPPRESSOR 1 (BES1) impairs transcription of genes concerned with brassinosteroid-mediated stimulation of hypo-cotyl extension[18]. Similarly, binding of UVR8 to MYB DOMAIN PROTEIN 73 (MYB73) and MYB77 inhibits transcription of genes involved in auxin-regulated lateral root formation[19], while UVR8 binding to MYB13 differentially affects transcription of auxin-responsive genes and fla-vonoid biosynthesis genes[31]. Furthermore, UVR8 has been shown to interact with a DNA methyltransferase to suppress DNA methylation[20].

Although the physical interaction of proteins with UVR8 is crucial in orchestrating UVR8-mediated responses, the mechanisms respon-sible for differential protein binding, which determine the nature of the response, are unknown. Hence, discovering these mechanisms is pivotal to understanding how the photoreceptor initiates diverse responses. In this study we show that UVR8 is phosphorylated in vivo at several sites. In particular, phosphorylation of a single amino acid in the C27 region, Serine 402 (S402), is stimulated by UV-B in a SPA-dependent manner. We provide evidence that S402 phosphorylation differentially affects protein interactions, strongly promoting binding of RUP proteins. S402 phosphorylation increases accumulation of proteins involved in UVR8 action by reducing their proteolytic degradation, and enhances the accumulation of phenylpropanoid and flavonoid compounds, which is a key response to UV-B radiation. Hence this study reveals a novel mechanism of UVR8 signalling and opens up new avenues of research to understand how UV-B radiation regulates plant growth and development.

## Results

### UVR8 is phosphorylated

Phosphorylation plays a key role in regulating plant photoreceptor function and photomorphogenesis[32–34]. Hence, we investigated whe-ther phosphorylation could regulate the interaction of proteins with UVR8. To test whether UVR8 is subject to phosphorylation in vivo, Arabidopsis seedlings expressing GFP-UVR8 were labelled with $^{32}$P and incorporation of radioactivity into immunoprecipitated GFP-UVR8 was visualised by autoradiography (Fig. 1a). Phosphorylation of GFP-UVR8 increased within 6 h exposure of seedlings to UV-B. No labelling was observed in control seedlings expressing GFP alone (Fig. 1a), demon-strating that phosphorylation was specific to UVR8. Quantification (see Supplementary Table 1) showed that net incorporation of $^{32}$P into UVR8 increased approximately 1.5-fold after 24 h UV-B exposure

(Fig. 1b). In vivo labelling of wild-type L*er* seedlings showed that native UVR8 is phosphorylated (Fig. 1c). Comparison of autoradiographs and western blots (Fig. 1a, c) reveals a slight mobility-shift of the labelled protein, consistent with phosphorylation.

To map the phosphorylation sites in UVR8 we used mass spec-trometry to identify phosphopeptides generated by trypsin digestion of immunoprecipitated GFP-UVR8. The analysis identified a relatively small number of phosphorylation sites. The key phosphopeptides and phosphorylated amino acids are shown in Fig. 1d and their presence in replicate tissue samples is presented in Supplementary Table 2. Within the C27 region, S402 was consistently identified as being phosphory-lated in replicate analyses with plants both exposed, or not, to UV-B. S402 is highly conserved among UVR8 sequences from diverse taxa (Supplementary Fig. 1). Several other possible sites were identified in C27 (T414, T417, S420, S421; Fig. 1d), but it is not clear which, if any, were phosphorylated.

We focused on phosphorylation of C27 because of its potential importance in protein interactions[15–23]. Plants expressing a UVR8 deletion mutant lacking C27 (GFP-ΔC27UVR8;[15]) had much reduced in vivo $^{32}$P-labelling, as did plants with an alanine mutation of S402 (GFP-UVR8$^{S402A}$; Fig. 1b, e), indicating that C27 is the major region of UVR8 phosphorylation and that S402 is the major site of phosphor-ylation within C27. Moreover, there was no effect of UV-B exposure on either mutant. Labelling of GFP-UVR8$^{S402A}$ was approximately 28% that of GFP-UVR8 in UV-B-treated seedlings (Fig. 1b, e), indicating that S402 is the principal site of UV-B-stimulated phosphorylation in UVR8.

To further study S402 phosphorylation we made an antibody that specifically detects UVR8$^{S402-P}$. To characterise the phospho-antibody, duplicate protein samples were run on the same gel and each half of the western blot was probed with either anti-GFP antibody or the anti-UVR8$^{S402-P}$ antibody (Fig. 2a). Whereas the anti-GFP antibody recog-nised GFP-UVR8 as a single band, the anti-UVR8$^{S402-P}$ antibody recog-nised two bands, a strong upper band, which increased in intensity in the sample from UV-B-treated plants, and a weak lower band, with the same mobility as the band recognised by the anti-GFP antibody. When the sample from UV-B-exposed plants was treated with lambda phos-phatase, the upper band disappeared (Supplementary Fig. 2a). The phospho-antibody recognised GFP-UVR8$^{S402A}$ mutant as only the lower band (Fig. 2a). Together these observations indicate that the upper band is the S402-phosphorylated form of UVR8. The mobility of the upper phosphorylated band is consistent with the band shift observed in the $^{32}$P-labelling experiments.

It was not possible to use the phospho-antibody directly to detect native UVR8 phosphorylation in total protein extracts of wild-type L*er* because the antibody recognises two non-specific bands with similar mobility to UVR8 (Supplementary Fig. 2b). To overcome this problem, UVR8 was immunoprecipitated with anti-UVR8 polyclonal antibody before being challenged with the anti-UVR8$^{S402-P}$ antibody. As shown in Fig. 2b, the upper band detected by the anti-UVR8$^{S402-P}$ antibody increased in intensity after UV-B exposure and was susceptible to lambda phosphatase. Whereas S402 phosphorylation clearly increased within 6 h of UV-B exposure, de-phosphorylation in minus-UV-B con-ditions was relatively slow, as an increased level was still evident after 6 h (Fig. 2c). Both dimeric and monomeric forms of UVR8 can be phosphorylated (Supplementary Fig. 2c), but the rate of de-phosphorylation is slower than the kinetics of re-association of mono-mers to form the dimer, which is complete within 1−2 h in vivo[28,35].

The phospho-antibody strongly recognises UVR8$^{S402-P}$ but also unphosphorylated UVR8, whereas the anti-GFP antibody and anti-UVR8 polyclonal antibody recognise only the lower band. The failure of the latter antibodies to detect the upper UVR8$^{S402-P}$ band indicates that UVR8$^{S402-P}$ is a small fraction of total UVR8; in routine western blots with anti-UVR8 antibodies, UVR8$^{S402-P}$ is undetectable.

The relatively low abundance of UVR8$^{S402-P}$ suggested that UV-B-stimulated phosphorylation might occur in the nucleus, since a small

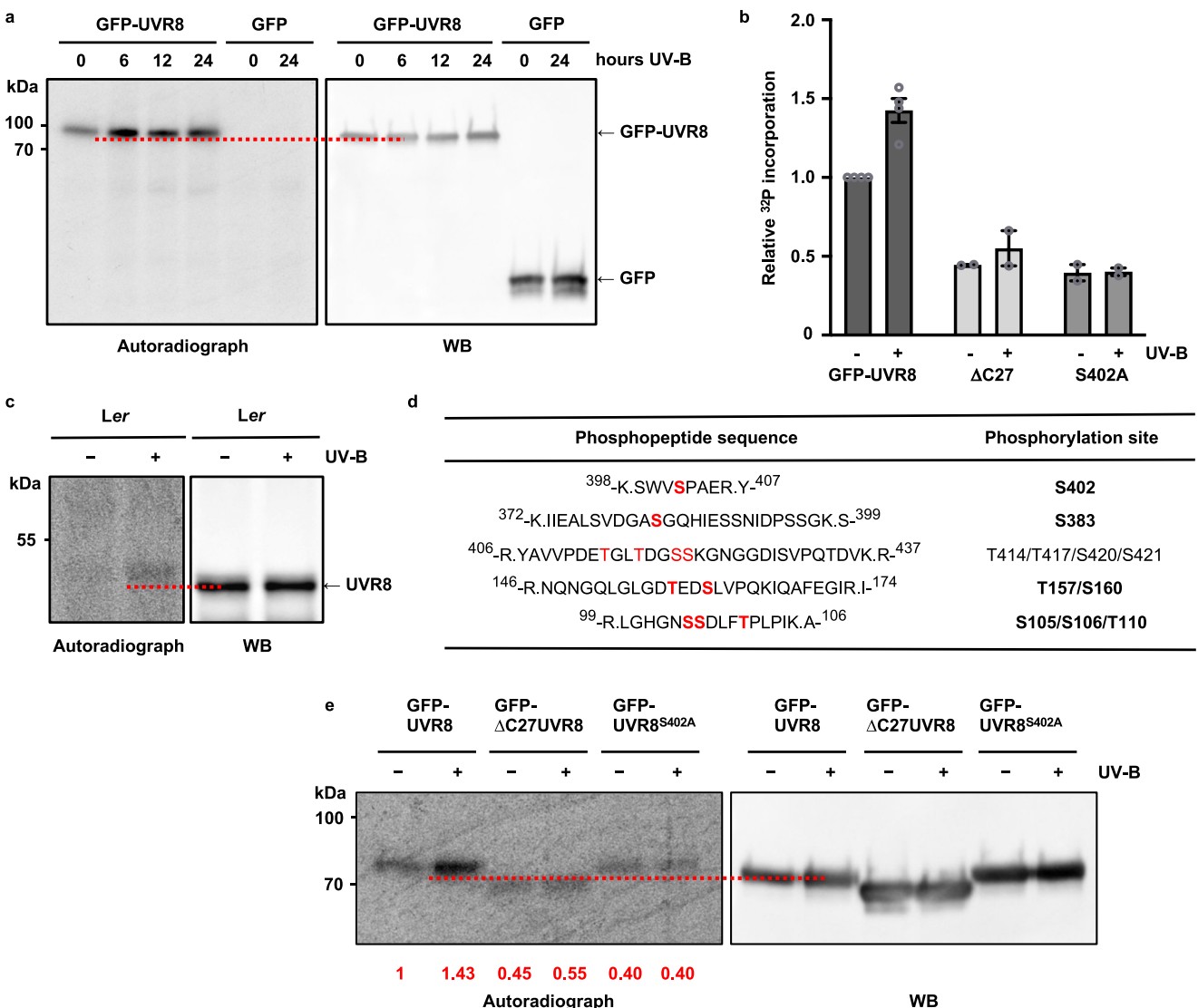

**Fig. 1 | UVR8 is phosphorylated. a** Arabidopsis *uvr8-1* seedlings expressing GFP-UVR8 or GFP grown in white light were labelled with $^{32}$P for 2 h and then exposed to supplementary 1.5 μmol m$^{-2}$ s$^{-1}$ UV-B. Protein extracts were incubated with anti-GFP beads. A western blot (WB) of immunoprecipitates was subjected to autoradiography to detect $^{32}$P and then probed with anti-GFP antibody. The dotted red line facilitates comparison of the mobility of bands detected by autoradiography and antibody. **b** Quantification of relative phosphorylation of GFP-UVR8, GFP-ΔC27UVR8 and GFP-UVR8$^{S402A}$ after 0 and 24 h UV-B exposure (see Supplementary Table 1). GFP-UVR8: mean ± S.E. (*n* = 4 biologically independent experiments); GFP-ΔC27UVR8 and GFP-UVR8$^{S402A}$: data points from *n* = 2 biologically independent experiments. **c**, L*er* seedlings labelled with $^{32}$P as in (**a**), then exposed to 3 μmol m$^{-2}$ s$^{-1}$ supplementary UV-B. Immunoprecipitates obtained using anti-UVR8 beads were examined as in (**a**), except with anti-UVR8 antibody. **d** The principal phosphopeptides identified by mass spectrometry (see Supplementary Table 2); confirmed phosphorylated amino acids are in bold. **e** Seedlings expressing GFP-UVR8, GFP-ΔC27UVR8, or GFP-UVR8$^{S402A}$ labelled with $^{32}$P, exposed or not to UV-B for 24 h, and analysed as in (**a**). Numbers below the lanes show the mean relative $^{32}$P incorporation.

proportion of cellular UVR8 accumulates in the nucleus following UV-B exposure[12–14]. To test this possibility we used plants expressing GFP-UVR8 targeted to the nucleus with a nuclear localisation signal (NLS)[12]. The intensity of the upper UVR8$^{S402-P}$ band recognised by the phospho-antibody was very strong relative to the lower UVR8 band in NLS-GFP-UVR8 plants, compared to GFP-UVR8 plants (Fig. 2d). Moreover, in contrast to GFP-UVR8 plants, there was no effect of UV-B treatment. To further examine the site of S402 phosphorylation we isolated cytosolic and nuclear fractions. The results show that nuclear UVR8 is strongly phosphorylated at S402 following UV-B exposure (Fig. 2e). Together these observations indicate that S402 phosphorylation increases as a consequence of UV-B-stimulated nuclear accumulation. The UV-B-stimulated formation of UVR8$^{S402-P}$ in the nucleus is important, because UVR8 acts in the nucleus to initiate responses through interactions with other proteins.

The C27 region of UVR8 contains a Valine-Proline (VP) motif (residues V410 and P411) that is critical for binding both COP1 and RUP proteins[16,21–23]. The amino acid sequence containing S402 is similar to the VP-domain in HY5 where Serine-36 is phosphorylated by SPA protein kinase activity (Fig. 2f)[34]. We therefore tested whether SPA proteins, which are nuclear localised[36], are involved in UVR8 phosphorylation. As shown in Fig. 2g, Arabidopsis plants mutated in all 4 SPA proteins (*spaQ* mutant[37]) failed to show an increase in abundance of the UVR8$^{S402-P}$ band following UV-B exposure, indicating that at least one SPA protein is required for UV-B-stimulated S402 phosphorylation. The response is retained in the *spa2-1,spa3-1,spa4-1* triple mutant[38], showing that SPA1 is sufficient for S402 phosphorylation. Furthermore, experiments with *spaQn* mutant plants expressing the kinase-dead R517E mutant of SPA1[39] show that UV-B-stimulated S402 phosphorylation is dependent on SPA1 kinase activity (Fig. 2g; Supplementary Fig. 2d).

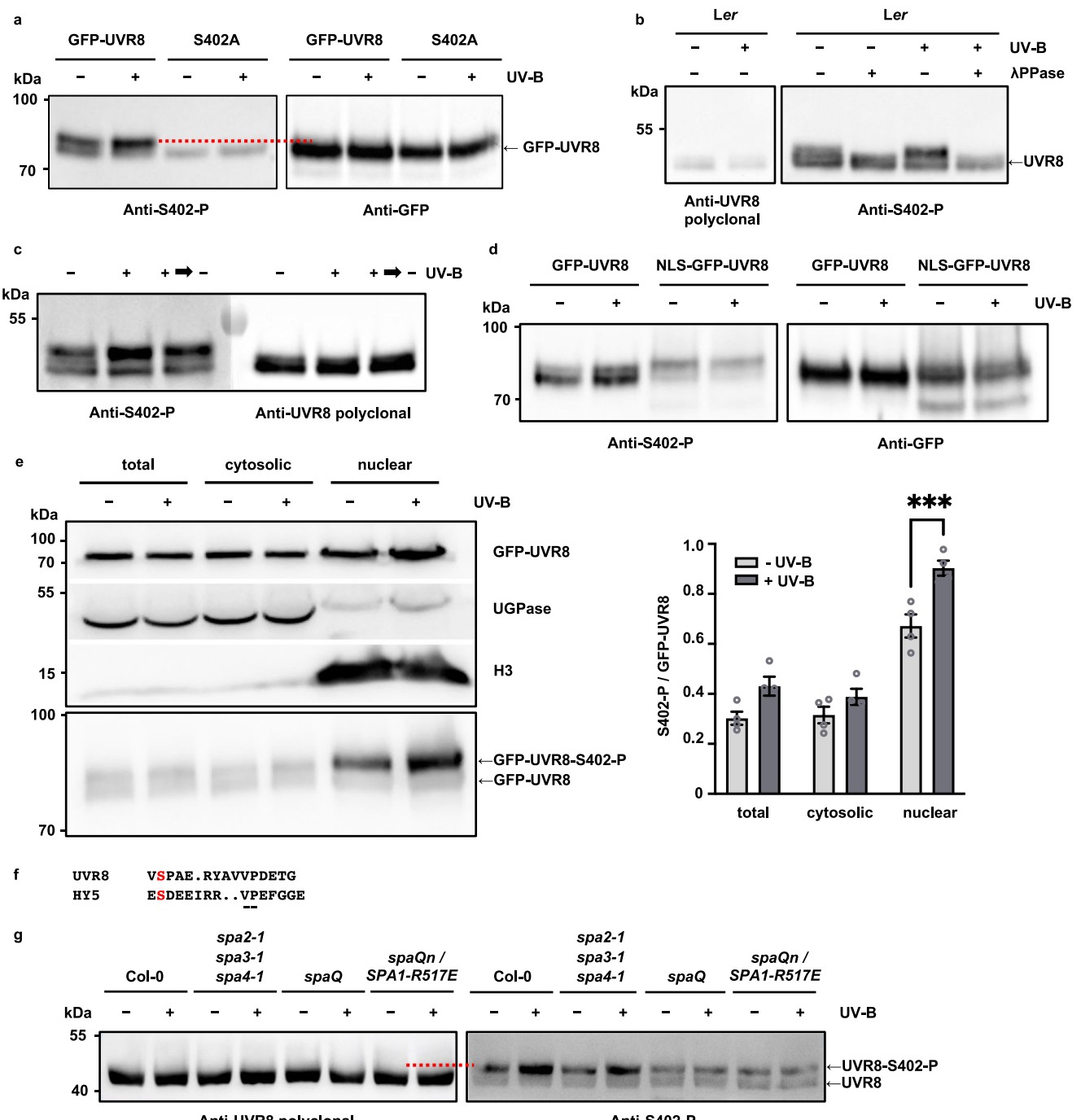

**Fig. 2 | Phosphorylation of UVR8 S402. a** 10-day old *uvr8-1* plants expressing GFP-UVR8 or GFP-UVR8[S402A] grown in white light (WL) were exposed to supplementary 1.5 μmol m⁻² s⁻¹ UV-B for 24 h. Western blot of duplicate protein samples was probed with anti-UVR8[S402-P] phospho-antibody or anti-GFP antibody. The dotted red line facilitates comparison of band mobilities. **b** 10-day old L*er* plants grown in WL were exposed to supplementary 3 μmol m⁻² s⁻¹ UV-B for 6 h. Immunoprecipitated UVR8 incubated, or not, with lambda protein phosphatase; western blot probed with anti-UVR8 polyclonal antibody or anti-UVR8[S402-P] phospho-antibody. **c** 10-day old L*er* plants grown in WL (-) were exposed to supplementary 3 μmol m⁻² s⁻¹ UV-B for 6 h (+) and then returned to WL for 6 h (+ → -). Western blot of duplicate immunoprecipitated UVR8 samples was probed with anti-UVR8 polyclonal antibody or anti-UVR8[S402-P] phospho-antibody. **d** 10-day old plants expressing GFP-UVR8 or NLS-GFP-UVR8 grown and exposed to UV-B as in (**a**). Western blot of protein samples probed with anti-UVR8[S402-P] phospho-antibody or anti-GFP antibody. **e** GFP-UVR8 plants exposed, or not, to UV-B as in (**a**) were used to make total, cytosolic and nuclear protein extracts. 3 horizontal strips from one western blot were probed with antibodies to detect GFP-UVR8, UGPase and histone H3; lower panel: a blot of duplicate samples was probed with anti-UVR8[S402-P] antibody. The panel to the right shows quantification of the S402-P band relative to the GFP-UVR8 band in each sample in *n* = 4 biologically independent experiments ±S.E.. The data were analysed using two-way ANOVA with Bonferroni's multiple comparisons test; UV-B has a significant effect on the relative band intensity in the nuclear fraction, *p* = 0.0005 (***). **f** UVR8 sequence containing S402 (in red) compared to the HY5 sequence containing S36. Conserved VP residues are underlined. **g** Plants of the genotypes indicated were grown and exposed to UV-B as in (**b**). UVR8 was immunoprecipitated and western blots of duplicate immunoprecipitated UVR8 samples were probed with anti-UVR8 polyclonal antibody (left panel) or anti-UVR8[S402-P] phospho-antibody (right panel). The panels are aligned using molecular mass markers; the dotted red line shows the position of the UVR8[S402-P] band. Representative blots are shown for *n* = 3 (**a**, **b**, **c** and **d**) or *n* = 4 (**e**, **g**) biologically independent experiments, each of which gave similar results.

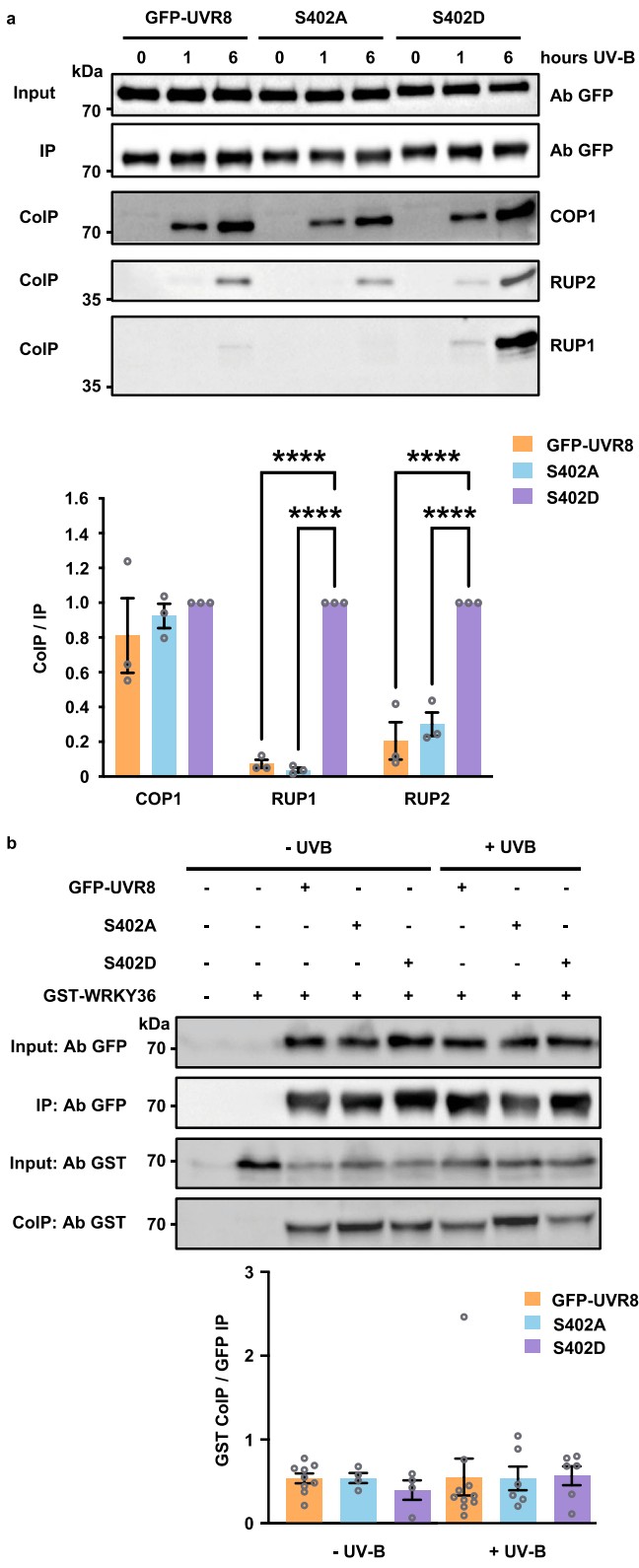

**Fig. 3 | S402 phosphorylation modifies protein interactions with UVR8. a** 14-day old *uvr8-1* plants expressing GFP-UVR8, GFP-UVR8[S402A] (S402A) or GFP-UVR8[S402D] (S402D), grown in white light were exposed to supplementary 3.0 μmol m$^{-2}$ s$^{-1}$ UV-B as indicated. Relative amounts of GFP-UVR8, S402A and S402D were assayed in protein extracts (Input) using anti-GFP antibody (Ab GFP). GFP-UVR8, S402A or S402D were immunoprecipitated and detected on western blots (IP) using anti-GFP antibody; co-immunoprecipitated (CoIP) COP1, RUP1 and RUP2 were detected using the corresponding antibodies. The lower panel shows quantification of the CoIPs of COP1, RUP1 and RUP2 relative to the IPs of GFP-UVR8, S402A or S402D after 6 h UV-B in *n* = 3 biologically independent experiments ±S.E.. The data were analysed using two-way ANOVA with Tukey's multiple comparisons test; the differences in CoIP/IP of RUP1 and RUP2 for S402D relative to GFP-UVR8 and S402A are significant, *p* < 0.0001 (****). **b** GFP-UVR8, S402A, S402D and GST-WRKY36 fusions were transiently expressed in *Nicotiana* leaves, subsequently exposed to supplementary UV-B for 3 h. Relative amounts of Input, IP and CoIP proteins were detected with the appropriate antibodies, as in (**a**). The bars in the lower panel correspond to lanes in the western blot, and show quantification of the CoIP of GST-WRKY36 relative to the IP of GFP-UVR8, S402A and S402D in *n* = 9 (GFP-UVR8 -UV-B), 4 (S402A -UV-B), 4 (S402D -UV-B), 10 (GFP-UVR8 + UV-B), 6 (S402A + UV-B) and 6 (S402D + UV-B) biologically independent experiments ±S.E.

phosphorylation, with those expressing the phosphomimetic mutant GFP-UVR8[S402D]. The charge added by the S402D mutation is demonstrated by the mobility shift in the protein (Fig. 3a Input). Lines with very similar levels of transgene expression were selected (Supplementary Fig. 3) so that results were not influenced by differences in the amount of photoreceptor. Interactions were tested by co-immunoprecipitation assays. Consistent with previous studies, interactions of both COP1[15,24,25,40] and RUP proteins[40] with GFP-UVR8 increased following exposure of plants to UV-B (Fig. 3a), and increased interaction was also seen in the S402 mutants. However, whereas no consistent difference was observed between GFP-UVR8, GFP-UVR8[S402A] and GFP-UVR8[S402D] in binding to COP1, GFP-UVR8[S402D] had strongly increased interaction with both RUP1 and RUP2 (Fig. 3a). Equivalent results were observed in several transgenic lines (Supplementary Fig. 4a). The increased RUP2 binding to GFP-UVR8[S402D] was also observed with proteins expressed transiently in *Nicotiana benthamiana* leaves (Supplementary Fig. 5a) and when proteins were expressed in mammalian cells (Supplementary Fig. 6). The results with the phosphomimetic mutant indicate that S402 phosphorylation promotes binding of RUP proteins to UVR8, possibly by enhancing electrostatic interaction between the proteins at the C27 binding site[23].

Increased binding of the RUP negative regulators is likely to impact UVR8 function. Since RUP proteins are known to promote re-dimerisation of UVR8 monomers[28,41], we examined whether S402 phosphorylation affects the ability of UVR8 to form dimers, to produce monomers following UV-B exposure, and the kinetics of re-dimerisation. However, no clear difference was observed between GFP-UVR8, GFP-UVR8[S402D] and GFP-UVR8[S402A] in these experiments (Supplementary Fig. 5b, c). Hence, any functional consequence of increased RUP binding caused by S402 phosphorylation may not be explained by an effect on UVR8 dimer/monomer status.

In addition, we monitored interaction of UVR8 with the WRKY36 transcription factor. GFP-UVR8 and GST-WRKY36 were co-transfected in *N. benthamiana* leaves and GFP-UVR8 was immunoprecipitated. The amount of co-immunoprecipitated GST-WRKY36 was independent of exposure to UV-B (Fig. 3b), consistent with previous findings[17]. There was no difference in the amount of GST-WRKY36 co-immunoprecipitated by GFP-UVR8, GFP-UVR8[S402A] and GFP-UVR8[S402D], indicating that S402 phosphorylation does not affect WRKY36 interaction with UVR8. Together the experiments with the S402 mutants provide evidence that S402 phosphorylation differentially affects protein interactions with UVR8: in particular, the phosphomimetic UVR8[S402D] mutation does not affect COP1 and WRKY36 binding, but strongly enhances binding of RUP proteins.

## Phosphorylation alters UVR8 interactions

Given the importance of the C-terminal region of UVR8 in making physical contact with other proteins[15–23], we examined whether phosphorylation of S402 modified interactions. Since at least 10 proteins are known to bind to UVR8, we decided to focus on interactions with COP1, RUP1 and RUP2 proteins and one of the interacting transcription factors, WRKY36. To test the role of S402 phosphorylation we compared transgenic lines expressing GFP-UVR8[S402A], which lacks S402

## Phosphorylation affects UVR8 responses

Since the HY5 transcription factor is a key effector of responses mediated by UVR8[24,26] we examined whether its expression is altered by S402 phosphorylation. HY5 protein accumulation was greater in GFP-UVR8[S402D] seedlings exposed to UV-B than in GFP-UVR8 seedlings, whereas GFP-UVR8[S402A] seedlings showed much lower HY5 accumulation (Fig. 4a). However, there was no corresponding change in the level of *HY5* transcripts (Fig. 4b), indicating that the effect of S402 phosphorylation on HY5 accumulation is mainly post-transcriptional. HY5 proteolysis is mediated by E3 ubiquitin-ligase complexes consisting of a CULLIN4 (CUL4)-DAMAGED DNA BINDING PROTEIN 1 (DDB1) scaffold bound to either COP1-SPA[24,25] or RUP proteins[30] as substrate receptors. Although S402 phosphorylation does not affect binding of COP1 to UVR8, our experiments suggest that it strongly increases binding of RUP1 and RUP2 (Fig. 3a), which would limit their availability to form an E3 ubiquitin-ligase and bind to substrates. Therefore, considering the above alternatives, reduced formation of the CUL4-DDB1-RUP1/RUP2 E3 ubiquitin-ligase is most likely to explain the increased stability of HY5 protein observed in GFP-UVR8[S402D] seedlings exposed to UV-B (Fig. 4a).

To further explore the effect of S402 phosphorylation on UVR8-mediated responses we examined accumulation of phenolic compounds in UV-B-acclimated plants growing under photoperiodic conditions. The formation of products derived from branches of the phenylpropanoid pathway, including flavonoids and hydroxycinnamic acid derivatives, is an important response to UV-B mediated by UVR8[2-6]. UV-B-acclimated GFP-UVR8[S402D] plants had increased accumulation of the key flavonoid biosynthesis enzyme chalcone synthase (CHS) compared to GFP-UVR8[S402A] (Fig. 4c; Supplementary Fig. 4b), but there was no corresponding change in *CHS* transcript levels (Fig. 4d). Treatment of plants with the proteasome inhibitor MG132 (which stimulated protein ubiquitylation; Supplementary Fig. 7) caused an increase in CHS accumulation (Fig. 4e), indicating that CHS abundance is modulated by proteasomal degradation, as reported previously[42]. Moreover, the increased abundance of CHS in GFP-UVR8[S402D] compared to GFP-UVR8[S402A] (Fig. 4c, e), was not evident in the presence of MG132 (Fig. 4e). Together the results suggest that S402 phosphorylation may stabilise CHS by inhibiting the activity of an unidentified E3 ubiquitin-ligase that targets CHS for proteolysis.

We previously reported that transfer of UV-B-acclimated plants to a 15-fold higher fluence rate of UV-B increased expression of several transcripts, including *CHS*[40]. Under similar conditions, S402D plants showed a 20-fold increase in *CHS* transcript level whereas S402A plants showed a significantly lower, 13-fold increase (Fig. 4f). Thus, S402 phosphorylation is able to regulate *CHS* expression via both transcript accumulation and protein stability, depending on the growth and treatment conditions.

Consistent with increased accumulation of CHS, UV-B-acclimated GFP-UVR8[S402D] plants had elevated levels of phenolic compounds in tissue extracts (Supplementary Fig. 8), including increased accumulation of flavonoids, notably kaempferol and quercitin glucosides, as well as sinapate glucosides (Fig. 4g; Supplementary Table 3); S402D plants had an approximately 20−25% increase in flavonoid/hydroxycinnamic acid content over S402A. These compounds have important roles in plants, including protection against damaging levels of UV radiation[2-6], abiotic stresses[43], and attack by pests and pathogens[44-46]. Furthermore, phenylpropanoid and flavonoid compounds are key nutrients in the harvested products of crops[47,48].

## Discussion

Recent research has shown that UVR8 interacts directly with multiple proteins to initiate specific responses. Moreover, the list of interactors continues to grow. However, the mechanisms that control differential protein binding, which are crucial in regulating and directing UVR8 activity, are unknown. The present study shows that UVR8 is phosphorylated in vivo and provides evidence that phosphorylation of a specific amino acid, S402, modulates protein interactions, protein stability and UVR8 mediated responses, as summarised in Fig. 5. UV-B exposure causes dissociation of UVR8 dimers, which initiates the accumulation of a small proportion of cellular UVR8 in the nucleus[12-14]. S402 phosphorylation is stimulated following UV-B exposure and is strongly enhanced in the nucleus, likely because it requires a nuclear localised SPA protein. We propose that SPA-dependent nuclear S402 phosphorylation occurs as a consequence of UV-B-stimulated nuclear accumulation. However, further research is needed to determine the kinetic relationship between monomer formation, nuclear accumulation and S402 phosphorylation, and whether de-phosphorylation occurs in the nucleus or cytosol. Although S402 is the major phosphorylation site of UVR8, UVR8[S402-P] is in low abundance in the cell and not readily detectable by standard anti-UVR8 antibodies. Nevertheless, we propose that S402 phosphorylation differentially modifies protein interactions with UVR8, strongly increasing the binding of RUP proteins (Fig. 5). In addition, S402 phosphorylation promotes the accumulation of specific proteins involved in mediating UVR8 responses by reducing their proteolytic degradation. Consequently, the altered protein accumulation enhances a key metabolic response to UV-B radiation, the biosynthesis of flavonoids and hydroxycinnamic acids.

Radio-labelling experiments show that the C27 region is the main site of UVR8 phosphorylation. The C27 region is involved in the interaction of several proteins with UVR8[15-23]. The topography, surface chemistry and charge presented by C27 amino acids will be crucial in binding specific proteins and phosphorylation will modify electrostatic interactions. COP1 and RUP2 interact with UVR8 via the C27 region and also through an interface of charged amino acids on the surface of the UVR8 β-propeller core[15,16,21-23]. The C27 VP motif is critical for binding both proteins[16,21-23]. The C27 S402A mutant is likely to retain some level of protein binding to the UVR8 core, which could reduce the impact of the mutation on interactions and responses relative to wild-type UVR8. However, the charge added by the phosphomimetic S402D mutation has the potential to strengthen interactions, which we see clearly with RUP proteins. The level of S402 phosphorylation of wild-type UVR8 is likely to be intermediate between S402A and S402D lines and will vary with growth and illumination conditions. Hence, comparison of S402A and S402D lines provides a reliable indication of the functional role of S402 phosphorylation. The present data indicate that phosphorylation of S402 in the C27 region enhances the interaction with RUP proteins but does not affect interaction with COP1 or WRKY36. Whether S402 phosphorylation influences binding of other interactors remains to be determined. In addition, the possible involvement of other phosphorylated amino acids in protein interactions can now be examined.

HY5 is an important effector of photomorphogenic responses and is subject to both transcriptional and post-translational regulation. Previously, it was reported that UV-B exposure stabilises HY5 because COP1-SPA dissociates from the CUL4-DDB1-COP1-SPA E3 ubiquitin-ligase that targets HY5 to form a complex with UVR8[25]. However, the observed stabilisation of HY5 in GFP-UVR8[S402D] seedlings exposed to UV-B cannot be explained by altered abundance of the CUL4-DDB1-COP1-SPA E3 ubiquitin-ligase because S402 phosphorylation does not alter COP1 binding to UVR8. In contrast, increased binding of RUP1 and RUP2 to UVR8 would clearly limit their availability to form the CUL4-DDB1-RUP1/RUP2 E3 ubiquitin-ligase that targets HY5 for proteolysis[30]. Hence, reduced availability of the RUP-E3 complex is the most likely explanation for the increased abundance of HY5 in GFP-UVR8[S402D] seedlings compared to GFP-UVR8[S402A] seedlings, and could impact the stability of other possible targets of the complex. A further potential consequence of enhanced RUP binding to UVR8[S402-P] is the increased availability of the CUL4-DDB1 scaffold to form other E3 ubiquitin-ligases, raising the possibility that UVR8 phosphorylation may impact processes beyond UV-B signalling. Moreover, it is possible that increased RUP binding to UVR8[S402-P] could affect interactions of UVR8

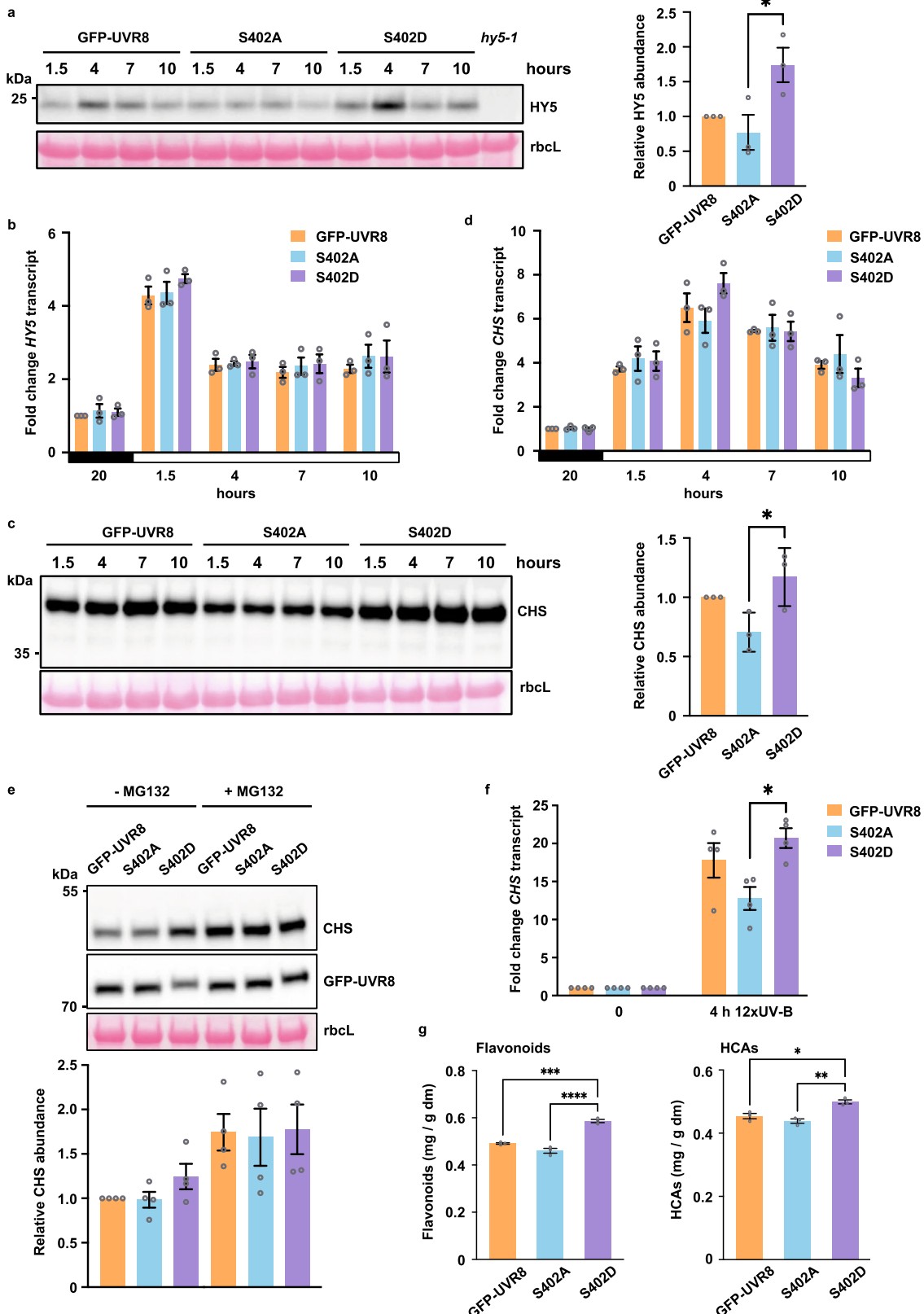

with other proteins, such as transcription factors, either through competition or enhanced recruitment.

In addition to its function in the COP1-SPA E3 ubiquitin-ligase that targets HY5 and various other proteins for degradation, SPA1 has been shown to have protein kinase activity[34,49]. Phosphorylation of HY5 at S36 by SPA1 kinase impairs its binding to COP1 and consequently

protects against targeted proteolysis[34]. The experiments with several *spa* mutants (Fig. 2g) show that UV-B stimulated S402 phosphorylation requires SPA1 kinase activity. The similarity of the amino acid sequences containing S402 and HY5 S36 is intriguing and raises the possibility that S402 could be phosphorylated directly by SPA protein kinase activity in the nucleus. Alternatively, it is possible that SPA

**Fig. 4 | S402 phosphorylation alters UVR8-mediated responses. a** *uvr8-1* seedlings expressing GFP-UVR8, GFP-UVR8[S402A] (S402A) or GFP-UVR8[S402D] (S402D) and control *hy5-1* seedlings were grown for 5 days in 16 h light/8 h dark in 100 μmol m$^{-2}$ s$^{-1}$ white light supplemented with 0.25 μmol m$^{-2}$ s$^{-1}$ UV-B. Western blot of protein samples was probed with anti-HY5 antibody; rbcL is a loading control; hours are numbered relative to the start (= 0) of the 16 h light period. Panel to the right shows relative quantification at 4 h; mean ± S.E. for $n = 3$ biologically independent experiments. The data were analysed using one-way ANOVA with Tukey's multiple comparisons test; the difference between S402A and S402D is significant, $p = 0.0349$ (*). **b** *HY5* transcript levels in seedlings as in (**a**); mean ± S.E. for $n = 3$ biologically independent experiments. Hours are numbered as in (**a**). **c** Plants grown as in (**a**) for 11 days; western blot probed with anti-CHS antibody. Panel to right shows relative quantification at 4 h; mean ± S.E. for $n = 3$ biologically independent experiments. The data were analysed using one-way ANOVA with Tukey's multiple comparisons test; the difference between S402A and S402D is significant, $p = 0.0361$ (*). **d** *CHS* transcript levels in plants as in (**c**); mean ± S.E. for $n = 3$ biologically independent experiments. **e** CHS accumulation assayed as in (**c**) in plants treated with MG132; tissue harvested 4 h into the light period. The bars in the lower panel correspond to lanes in the western blot and show quantification; mean ± S.E. for $n = 4$ biologically independent experiments. **f** *CHS* transcripts in plants grown as in (**c**), but transferred to 3 μmol m$^{-2}$ s$^{-1}$ UV-B for 4 h from the start of the light period; mean ± S.E. for $n = 4$ biologically independent experiments. The data were analysed using one-way ANOVA with Tukey's multiple comparisons test; the difference between S402A and S402D is significant, $p = 0.0259$ (*). **g** Total amounts of flavonoids and hydroxycinnamic acids (HCAs) in plants grown as in (**a**) for 20 days (details in Supplementary Table 3); mean ± S.E. for $n = 3$ biologically independent experiments. The data were analysed using one-way ANOVA with Tukey's multiple comparisons test; the differences between S402D and GFP-UVR8 and S402D and S402A are significant; for flavonoids $p = 0.0002$ (***) and $p < 0.0001$ (****) respectively; for HCAs $p = 0.0109$ (*) and $p = 0.0025$ (**) respectively.

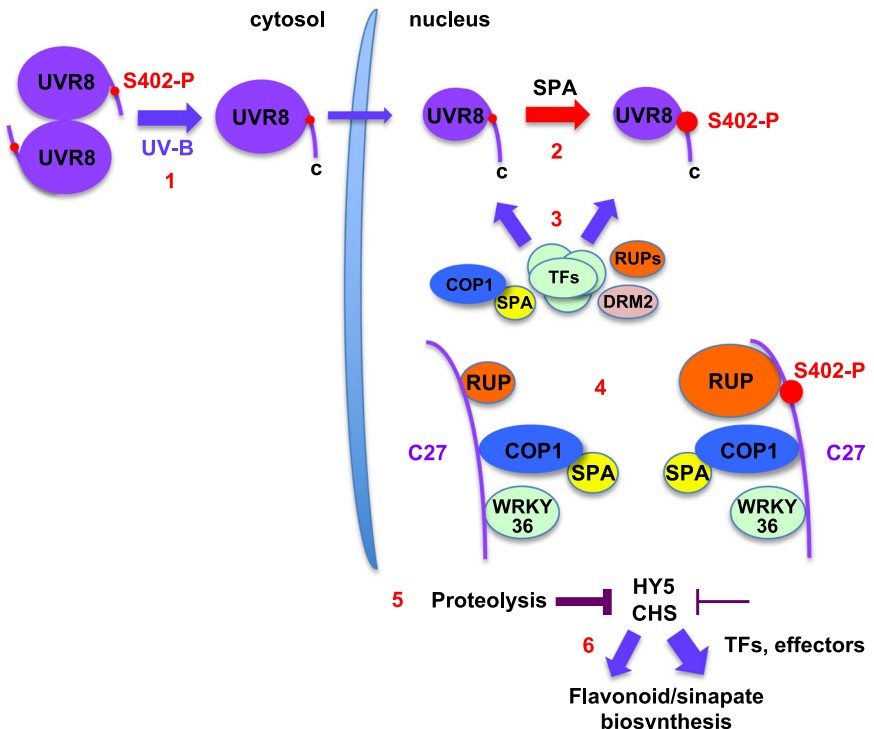

**Fig. 5 | Model summarising the impact of nuclear UVR8 S402 phosphorylation.** (1) Cytosolic UVR8 has detectable S402 phosphorylation. UV-B photoreception by UVR8 leads to accumulation of a small fraction of UVR8 in the nucleus where S402 phosphorylation is enhanced (represented by the increased size of S402 in the schematic) (2) in a process requiring a SPA protein. (3) Multiple proteins (COP1-SPA, RUP1, RUP2, several transcription factors and DRM2 methyltransferase) interact with UVR8 monomers in the nucleus, in part via the C27 region. (4) S402 phosphorylation promotes RUP1 and RUP2 protein binding (represented by the increased size of RUP), but does not affect COP1 or WRKY36 binding to UVR8. (5) S402 phosphorylation enhances accumulation of HY5 and CHS proteins by decreasing proteolysis; for HY5 this may be due to reduced availability of CUL4-DDB1-RUP1/RUP2 E3 ubiquitin-ligase. (6) S402 phosphorylation increases biosynthesis of flavonoids and sinapate glucosides, which involves multiple transcription factors, effectors and enzymes, including CHS.

kinase phosphorylates another protein that phosphorylates UVR8. Residual S402 phosphorylation is observed in *spaQ* plants in the absence of UV-B, indicating that an additional, unidentified kinase can phosphorylate S402 under these conditions. Nevertheless, although the regulation of S402 phosphorylation is not fully understood, our findings reveal both a novel role for SPA proteins in regulating UVR8 action and an additional way in which SPAs regulate HY5 accumulation, in that SPA-dependent S402 phosphorylation contributes to HY5 stabilisation.

Altered protein interactions with UVR8 are likely to modify downstream responses. However, responses to UV-B involve the complex interplay of numerous effectors, and since S402 phosphorylation could affect interaction of UVR8 with multiple proteins, probably including some not yet identified, it is not yet possible to explain precisely how it leads to altered responses. The complexity of responses is exemplified by phenolic biosynthesis, which involves multiple enzymes and transcriptional regulators. S402 phosphorylation increases CHS stability, but the underlying mechanism is not clear. It is known that a Kelch domain-containing F-box protein (KFB[CHS]) is involved in CHS proteolysis and its expression is reduced following UV-B exposure[42], but whether S402 phosphorylation affects the F-box protein is unknown. Nevertheless, the data presented here for the phosphomimetic UVR8[S402D] mutant indicate that S402 phosphorylation stimulates the accumulation of flavonoids and hydroxycinnamic acids that are important in plant protection and enhance nutritional quality of plant products.

In conclusion, discovering the mechanisms of UVR8 action is key to understanding how diverse plant species respond to UV radiation. These responses underpin adaptation and acclimation in natural environments and enhance desirable traits in various crops. The present study highlights the role of S402 phosphorylation in modulating responses to UV-B radiation in Arabidopsis. Moreover, since S402 is highly conserved among UVR8 sequences, S402 phosphorylation is likely to be important in a wide range of plant species. Hence this study provides a basis to understand how phosphorylation of S402 and additional UVR8 amino acids impacts the binding of multiple interactors that regulate responses to UV-B.

## Methods

### Plant materials and light treatments

Seeds of wild-type L*er* and Col-0 were obtained from Nottingham Arabidopsis Stock Centre. Seeds of the *uvr8-1* mutant[50] were provided by Dr Dan Kliebenstein (University of California, Davis). Seeds of *spaQ* (*spa1-3 spa2-1 spa3-1 spa4-1*)[37]; *spa2-1 spa3-1 spa4-1*[38]; and 2 transgenic lines (56-21 and 57-2) of *spaQn* expressing SPA1[R517E][39] were produced in Dr Ute Hoecker's laboratory (University of Köln). The transgenic line expressing the CaMV *35 S* gene promoter::GFP fusion in the L*er* background was obtained from Dr Robert Sablowski (John Innes Centre, UK).

Homozygous transgenic lines expressing GFP-UVR8, GFP-ΔC27UVR8, NLS-GFP-UVR8, GFP-UVR8[S402A] and GFP-UVR8[S402D] driven by the CaMV *35 S* promoter were made in the *uvr8-1* background by Agrobacterium-mediated transformation. Fusions were cloned into the pEZR(K)L-C vector[12,15,26] at the *Eco*RI and *Xma*I restriction sites using the In-Fusion HD Cloning kit (Takara Bio, USA). Cloned DNA sequences were validated by commercial sequencing. Several independent T3 lines were obtained for each fusion and lines were selected for experiments based on their similar levels of transgene expression (Supplementary Fig. 3). The experiments were undertaken with at least 2 independent lines for each fusion: GFP-UVR8 lines T3-26-5/11 and T3-11-4; GFP-UVR8[S402A] lines T3-52-6/13 and T3-51-8; GFP-UVR8[S402D] lines T3-24-12/14/15 and T3-17-2. Supplementary Fig. 4 shows key experiments (protein interactions; CHS protein accumulation) with several transgenic lines, demonstrating that the different lines produce equivalent results.

Plants were grown on compost or on agar plates containing half-strength Murashige and Skoog salts (½ MS) in either a 16 h/8 h light/dark cycle or constant white light (100 μmol m$^{-2}$ s$^{-1}$) provided by warm-white LEDs in a Micro-Clima growth cabinet (Snijders Labs, Netherlands) at 22 °C. UV-B was provided by a narrowband fluorescent source (Philips TL20W/01RS tubes; wavelength maximum 312 nm; spectrum shown in[15]). White light fluence rates were measured using a LI-250A light meter attached to a LI-190 quantum sensor (LI-COR, Lincoln, USA). UV-B fluence rates were measured using a Spectro Sense 2 SKL904 meter and a UV-B sensor, SKU 430/SS2 (Skye Instruments, Powys, UK).

### Tissue fractionation

Homozygous T3 GFP-UVR8 plants were grown on ½ MS agar plates for 14 days in constant 100 μmol m$^{-2}$ s$^{-1}$ white light. Plants were exposed, or not, to 1.5 μmol m$^{-2}$ s$^{-1}$ narrowband UV-B for 24 h. For each plant sample, 3 g tissue was treated with 1% formaldehyde for 15 min as described[51] to reduce loss of UVR8 from nuclei during fractionation. Cross-linking was stopped by adding glycine to a final concentration of 0.125 M and tissue was washed 3 times to remove formaldehyde. Tissue was frozen in liquid nitrogen and used for preparation of cytosolic and nuclear fractions using a published method[14]. Tissue was ground in 10 ml of lysis buffer (20 mM Tris pH 7.4, 25% (v/v) glycerol, 150 mM NaCl, 2 mM EDTA, 2.5 mM MgCl$_2$, 250 mM sucrose, 1 mM dithiothreitol, 1 mM PMSF) and homogenised by gentle rotation for 10 min at 4 °C. The homogenate was filtered through 3 layers of miracloth

(475855, Millipore) and the flow-through taken as the 'total' tissue sample. The homogenate was centrifuged at 1500 × *g* at 4 °C for 10 min and the supernatant collected (the 'cytosolic' fraction). The pellet was gently resuspended in 5 ml NRBT buffer (20 mM Tris pH 7.4, 25% (v/v) glycerol, 2.5 mM MgCl$_2$, 0.2% (v/v) Triton X-100), washed 3 times by centrifuging as above, and washed once with NRBT buffer minus Triton X-100. The nuclear pellet was finally resuspended in 80 μl of 4X SDS sample buffer (the 'nuclear' fraction).

Aliquots of the total tissue sample, cytosolic and nuclear fractions were used for SDS-PAGE and immunodetection. From the 3 g of starting material/10 ml lysis buffer used for each plant sample (plus or minus UV-B treatment), 30 μl of the total sample, 30 μl of the cytosolic fraction and 40 μl of the nuclear fraction was loaded on each lane of the gel used to immunodetect GFP-UVR8, cytosolic UGPase and nuclear histone H3; the western blot was divided horizontally into 3 strips, each incubated with the appropriate antibody. A second gel was loaded with duplicate samples of the same volumes and used to immunodetect UVR8[S402-P].

### In vivo $^{32}$P labelling

Seedlings were grown on 2.5 cm diameter filter paper discs on ½ MS agar plates for 6 days in constant white light. Discs were removed, blotted dry with tissue and placed in a cell culture plate. An aliquot of $^{32}$P-orthophosphate (specific activity 8500-9120 Ci/mMole; PerkinElmer, USA) was diluted with ½ MS liquid medium and 300 μl added to each filter paper disc. The seedlings were left for 2 h in white light before being exposed to supplementary UV-B. Total protein was extracted from the seedlings and used for immunoprecipitation of UVR8, as described below. Immunoprecipitated protein was run on SDS-PAGE and the presence of $^{32}$P on the western blot was detected by autoradiography. UVR8 was immunodetected on the western blot using the appropriate antibody. Quantification of relative $^{32}$P-labelling of UVR8 bands was undertaken using ImageJ software to measure the intensity of the bands obtained by western immunodetection and autoradiography. The amount of radioactivity taken up by the seedlings and present in the protein extracts was determined by measuring the total volume of the extract and the radioactivity (in cpm) in 1 μl extract using scintillation counting. The relative values obtained for the phosphorylated UVR8 band intensity, detected by autoradiography, were normalised for differences in the amount of radioactivity taken up by the seedlings and also for differences in the amount of immunoprecipitated UVR8, as detected on the western blot (Supplementary Table 1).

### Inhibition of proteolysis

Plants were grown on ½ MS agar plates under a 16 h/8 h light/dark cycle in 100 μmol m$^{-2}$ s$^{-1}$ white light supplemented with 0.25 ± 0.05 μmol m$^{-2}$ s$^{-1}$ narrowband UV-B for 12–14 days (UV-B-acclimated plants[40]). Eleven hours before the start of the light period plants were placed in ½ MS liquid medium containing 1% DMSO and 100 μM MG132 (or not in the control)[35]. Tissue was harvested 4 h after the start of the light period.

### Protein methods

Total protein was extracted from plant tissue in microextraction buffer (20 mM HEPES, pH 7.8, 450 mM NaCl, 50 mM NaF, 200 μM EDTA, 500 μM PMSF, 1 mM DTT, 25% glycerol, with one protease inhibitor tablet (Complete Mini, Roche) added per 10 ml)[12]. Standard protocols were used for SDS-PAGE and immunodetection on western blots[12,40]. Imaging of western blots was undertaken with a Fusion FX system (Vilber, Collégien, France). The antibodies used for immunodetection were an anti-GFP antibody (from either Chromotek, Germany, Cat. No. 3h9, or Clontech, Saint-Germain-en-Laye, France, Cat. No. 632375), an anti-UVR8 polyclonal antibody[52], anti-COP1 antibody[53], anti-HY5 antibody (Agrisera, Vännäs, Sweden, Cat. No. AS12 1867), anti-ubiquitin

antibody (Agrisera, Vännäs, Sweden, Cat. No. AS08 307), anti-CHS antibody (Santa Cruz Biotechnology, Heidelberg, Germany, Cat. No. sc-12620), anti-HA antibody (Roche, Basel, Switzerland, Cat. No. 3F10), anti-GST antibody (GenScript, Oxford, UK, Cat No. A00865), anti-bodies specific to RUP1 or RUP2[40] or anti-UVR8$^{S402-P}$ phospho-antibody. The latter antibody was a polyclonal obtained by double affinity pur-ification using phospho-modified and non-modified peptides CGKSWVS(PO3H2)PAERYA (Eurogentec, Belgium).

For protein phosphatase treatment, protein samples were incu-bated with 400 units of lambda phosphatase (New England Biolabs, Hitchin, UK) at 37 °C for 30 min.

Immunoprecipitation of GFP-UVR8 followed a published protocol[15,40] with the GFP-Trap system (Chromotek, Germany; Cat. No. gta-20). Immunoprecipitation of native UVR8 was undertaken using Dynabeads Protein A (ThermoFisher, Cat. No. 10002D) conjugated to UVR8 polyclonal antibody at 4 °C with rotation for 30 min. The antibody-Protein A-Dynabeads complexes were further washed prior to use for immunoprecipitation.

Quantification of protein bands was undertaken using ImageJ software, normalising the signal relative to the Rubisco large subunit (rbcL) loading control.

### Transient expression in *N. benthamiana*

DNA sequences required to produce the protein fusions used in this study were amplified by PCR using the CloneAmp HiFi PCR premix (Takara Bio, USA) following the manufacturer's instructions. The fusions used for transient expression were cloned into the pEZR(K)L-C vector at appropriate restriction sites using the In-Fusion HD Cloning kit (Takara Bio, USA). The cloned DNA sequences were validated by commercial sequencing. Purified plasmid DNA was used to transform electro-competent *Agrobacterium* GV3101 cells.

*N. benthamiana* leaves were transfected with *Agrobacterium* containing the designated plasmids[54]. *Agrobacterium* cells were sus-pended in infiltration buffer (10 mM MgCl$_2$, 10 mM MES, 200 μM acetosyringone, pH 5.6) and injected into the lower epidermis of the leaf. Following infiltration, plants were returned to the growth cham-ber in white light for 60 h to allow expression of transfected proteins. Where indicated, plants were exposed to white light supplemented with 3 μmol m$^{-2}$ s$^{-1}$ narrowband UV-B light for 3 h prior to protein extraction.

### Expression in mammalian cells

Human embryonic kidney (HEK) 293 T cells were routinely cultured in Dulbecco's modified Eagle's medium supplemented with 10% heat-inactivated fetal bovine serum, penicillin/streptomycin and Normocin (InvivoGen, Toulouse, France) in a humidified 5% (v/v) CO$_2$ atmosphere at 37 °C[55]. Approximately 2 million HEK293T cells were plated into 100 mm dishes at least 24 h before transfection. The coding sequences of Arabidopsis UVR8 wild-type and mutants N-terminally fused with GFP, and RUP1, RUP2 or COP1 N-terminally fused with HA were cloned into the pcDNA3.1 mammalian expression plasmid (ThermoFisher) between the *Hind*III and *Eco*RI sites using an In-Fusion HD Cloning kit (Takara Bio, USA). For the transfection, 5 μg of total plasmid DNA (including 2.5 μg of a GFP-UVR8 fusion and 2.5 μg of a HA-RUP/COP1 fusion) was mixed with 30 μg polyethylenimine (PEI) in a final volume of 500 μl of 150 mM NaCl. The DNA-PEI mixture was mixed and incu-bated at room temperature for 10 min before adding dropwise to the cells[55]. Transfected cells were cultured for 48 h to allow for optimal expression of the transfected constructs. Cells were exposed, or not, to 0.1 μmol m$^{-2}$ s$^{-1}$ narrowband UV-B for 4 h and then washed once with phosphate buffered saline. The cell pellet was extracted with lysis buffer (150 mM NaCl, 1% Triton X-100, 50 mM Tris HCl (pH 8.0), 0.5 mM PMSF, EDTA-free Protease Inhibitor Cocktail (1 tablet per 10 mL lysis buffer, Roche), PhosSTOP phosphatase inhibitor (1 tablet per 10 mL lysis buffer, Roche)). The extracts were incubated on ice for

20 min and then subjected to liquid Nitrogen freeze-thaw on ice for 20 min, followed by centrifugation at 20,000 x *g* for 15 min to obtain the supernatant. The same volume of each extract was incubated with 25 μl GFP-Trap beads and incubated at 4 °C for 30 min. The protein-beads complex was washed with lysis buffer (without PMSF, Protease inhibitor and PhosSTOP) 3 times and then boiled at 95 °C for 5 min with 2x SDS sample buffer (120 mM Tris/Cl pH 6.8, 20% glycerol, 4% SDS, 0.04% bromophenol blue, 10% β-mercaptoethanol). Samples were run on SDS-PAGE for immunodetection of proteins on western blots.

### Phosphorylation site mapping

Plants expressing GFP-UVR8 were grown for 10 days in constant white light and then exposed, or not, to supplementary 1.5 μmol m$^{-2}$ s$^{-1}$ UV-B for 24 h. Three independent biological replicates were used for the plus and minus UV-B treatments. GFP-UVR8 was immunoprecipitated from protein extracts and run on SDS-PAGE. Bands containing the protein were excised from the gel (6 bands per replicate) and washed for 15 min sequentially with 200 μL of Milli-Q water, 200 μL of acet-onitrile, 200 μL of 100 mM ammonium bicarbonate, 200 μL 100 mM ammonium bicarbonate/acetonitrile (50:50 v/v) and finally with 100 μL of acetonitrile for 10 min and briefly dried in a speed-Vac. Cysteine reduction and alkylation was achieved by adding 50 μl of 10 mM dithiothreitol followed by incubation at 56 °C for 1 h, then centrifugation to discard the supernatant. Alkylation was performed by adding 50 μL of 50 mM iodoacetamide in 20 mM ammonium bicarbonate and incubation on a shaker at room temperature in darkness for 30 min. The sample was centrifuged, and resultant supernatant discarded. In-gel digestion was carried out with trypsin (12 μg/mL in 20 mM ammonium bicarbonate) and incubation at 30 °C for 16 h on a shaking incubator. Acetonitrile (equal volume used to cover gel pieces as above) was added and peptide mixtures were extracted by shaking at 30 °C for 15 min. The resulting supernatant was transferred to a fresh tube. Peptides in the gel pieces were further extracted first by adding of 5% formic acid and shaking for 15 min, then by adding the same volume of 100% acetonitrile and shaking for another 45 min. The supernatant was then collected and transferred to the first tube. The gel pieces were finally washed with acetonitrile for 10 min and the three pooled fractions were combined with other replicate IP fractions to generate 6 pooled IP samples (WL1-3 and UV-B 1-3) then dried in a speed-Vac and stored at −20 °C until further processing.

Pooled IP samples were reconstituted in 50 μl 1% formic acid and 15 μl aliquots of each was analysed on a Q-Exactive Plus mass spec-trometer interfaced with a 3000 RSLC Nano liquid chromatography system. Pooled IP sample was loaded on to a 2 cm trapping column (PepMap) at 10 μl/min flow rate using a loading pump and analyzed on a 50 cm analytical column (EASY-Spray column, 50 cm 75 mm ID) at 300 nl/min flow rate. LC buffers used were the following: buffer A (0.1% formic acid in Milli-Q water (v/v)) and buffer B 80% acetonitrile and 0.1% formic acid in Milli-Q water (v/v). Aliquots of 15 μl were loaded at 10 μL/min onto a trap column which was equilibrated with 0.1% tri-fluoroacetic acid. The trap column was washed for 3 min and then the trap column was switched in-line with the analytical column main-tained at constant temperature of 50 °C. The peptides were eluted from the column with a linear gradient from 2% to 35% B over 120 min. Q-Exactive Plus was operated in data dependent positive ion mode. The source voltage was set to 2.0 kV and the capillary temperature was 250 °C. A scan cycle comprised MS1 scan (m/z range from 350 to 1600, ion injection time of 20 ms, resolution 70,000 and automatic gain control (AGC) 1e6) acquired in profile mode, followed by 15 sequential dependent MS2 scans (resolution 17,500) of the most intense ions fulfilling predefined selection criteria (AGC 2e5, maximum ion injec-tion time 100 ms, isolation window of 1.4 m/z, fixed first mass of 100 m/z, spectrum data type: centroid, underfill ratio 1%, exclusion of unassigned, singly and >7 charged precursors, peptide match

preferred, exclude isotopes on, dynamic exclusion time 45 s). The HCD collision energy was set to 27% of the normalised collision energy. Mass accuracy was checked before the start of samples analysis.

Raw ms files were analysed using Proteome Discoverer version 2.2.0.388 with Mascot (Matrix Science) as the search engine using the following parameters: Mascot expectation value less than or equal to 0.05 with high peptide confidence and minimum of 1 peptide per protein. Mascot search criteria: Protein Database: Sprot (Arabidopsis thaliana), fragment mass tolerance (0.06 Da), precursor mass tolerance (10 ppm), Dynamic modifications: Acetyl (N-term), Dioxidation (M), Oxidation (M), Gln->pyro-Glu (N-term Q), Deamidated (NQ), Phospho (ST) and Phospho (Y). Static modification: Carbamidomethyl (C). Individual phosphopeptides with ion scores above the Mascot Ion Score threshold were manually annotated to localise the site of phosphorylation using the mascot delta score.

### Transcript assays

The abundance of *HY5* and *CHS* transcripts was assayed relative to control *ACTIN2* transcripts by reverse transcriptase-qPCR. Plants were grown on ½ MS agar plates under a 16 h/8 h light/dark cycle in $100 \, \mu mol \, m^{-2} \, s^{-1}$ white light supplemented with $0.25 \pm 0.05 \, \mu mol \, m^{-2} \, s^{-1}$ narrowband UV-B (UV-B-acclimated plants[40]) for either 5 or 11 days to assay *HY5* and *CHS* transcripts respectively. Total RNA was extracted using the RNeasy plant mini kit (Qiagen). First-strand complementary DNA was then synthesised from $1 \, \mu g$ of the extracted total RNA with QuantiTect Reverse Transcription kit (Qiagen). The quantitative qPCR was performed using Brilliant III Ultra Fast SYBR qPCR Master Mix (Agilent Technologies) on a StepOnePlus Real-Time PCR System (Applied Biosystems, Life Technologies) according to the manufacturer's instructions. The primer sequences used were: *HY5*: forward GGCTGAAGAGGTTGTTGAGGAAC, reverse AGCATCTGGTTCTCGTTCTGAAGA; *CHS*: forward CAGACAGGACATCGTGGTGGT, reverse ACATGAGTGATCTTTGACTTGG; *ACTIN2*: forward GTATTGTGCTGGATTCTGGTG, reverse GAGGTAATCAGTAAGGTCACG. Data shown are from three independent biological repeats and two or three technical replicates.

### Analysis of phenolic compounds

Plants were grown under a 16 h/8 h light/dark cycle in $100 \, \mu mol \, m^{-2} \, s^{-1}$ white light supplemented with $0.25 \pm 0.05 \, \mu mol \, m^{-2} \, s^{-1}$ narrowband UV-B for 14 days (UV-B-acclimated plants[40]). For measurement of the absorption spectrum of tissue extracts, whole plants grown on ½ MS agar plates were harvested 4 h after commencement of the light period and homogenised in $400 \, \mu l$ 80% (v/v) methanol. Homogenised samples were incubated for 15 min at 70 °C then centrifuged for 10 min. Supernatants were vacuum-dried at 65 °C and the dried material dissolved in $1 \, \mu l$ 80% methanol mg$^{-1}$ fresh weight. $5 \, \mu l$ of each methanolic extract was added to $995 \, \mu l$ 80% methanol prior to measuring the absorption spectrum.

Identification and quantification of phenolics by HPLC was based on a published method[56]. Plants were grown as above, but on compost for 20 days rather than plates. In brief, approximately 40 mg of freeze-dried tissue samples were extracted in duplicate with $1500 \, \mu l$ of 60% methanol, 39.5% water and 0.5% formic acid (v/v/v) in a thermoshaker (Eppendorf, Hamburg, Germany) operating at 1400 rpm at 20 °C for 40 min. Subsequently, samples were centrifuged at $9250 \times g$ at 20 °C for 5 min. The supernatant was collected in a 5 mL volumetric flask and the pellet was redissolved in $1500 \, \mu l$ of 60% methanol, 39.5% water and 0.5% formic acid (v/v/v) and re-extracted in the thermoshaker as above at 20 °C for 15 min, before centrifuging again. This procedure was repeated once, resulting in a total of three extraction steps. The combined supernatants were filled up to 5 mL and filtered with 0.2 μm regenerated cellulose membrane (Chromafil Xtra, MACHEREY-NAGEL GmbH & Co. KG, Düren, Germany). The filtrate was used for HPLC analyses.

Phenolic compounds including hydroxycinnamic acid derivatives and flavonoid glycosides were quantified[56] using a Shimadzu Prominence HPLC (Shimadzu, Kyoto, Japan) equipped with a DGU-20A5 degasser, LC-20AT pump, SIL-20AC autosampler, CTO-10AS column oven, and SPD-M20A photodiode array detector. An Ascentis Express F5 column (150 × 4.6 mm i.d., 5 μm particle size, Merck, Darmstadt, Germany) protected by an Ascentis F5 guard column (5 × 4.6 mm i.d., 5 μm particle size, Merck) was used at 25 °C. Eluent A was 0.5% acetic acid, and eluent B was 100% acetonitrile. The gradient (eluent B) was 5–12% (0–3 min), 12–25% (3–46 min), 25–90% (46–49.5 min), 90% isocratic (49.5–52 min), 90–5% (52–52.7 min), and 5% isocratic (52.7–59 min) at a flow rate of 0.85 ml * min$^{-1}$. Phenolic compounds were measured at wavelengths of 320 nm for hydroxycinnamic acid derivatives and 370 nm for flavonoid glycosides. For quantification authentic standards of quercetin-3-glucoside, kaempferol-3-glucoside, and chlorogenic acid were used (Roth, Karlsruhe, Germany).

### Statistical analysis

Where representative data are presented, the experiments were repeated at least 3 times. Significant differences between mean values were tested using statistical analyses in GraphPad Prism 9 software, as stated in the Figure legends.

### Reporting summary

Further information on research design is available in the Nature Portfolio Reporting Summary linked to this article.

## Data availability

The data that support the findings of this study are available within the paper and its Supplementary Information files (uncropped gel images and blots; source data for graphs). The mass spectrometry proteomics data have been deposited to the ProteomeXchange Consortium via the PRIDE partner repository with the dataset identifier PXD035649. Source data are provided with this paper.

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

## Acknowledgements

The research was funded by a UK Biotechnology and Biological Sciences Research Council grant (BB/T003251/1) to G.I.J. and J.M.C., which supported W.L. and A.H.. W.L. was additionally supported by a China Scholarship Council PhD studentship. G.G. is supported by a Sainsbury PhD Studentship from the Gatsby Charitable Foundation. We thank Konstantina Malengou for technical assistance, Dr Brian Hudson and Bethany Dibnah for providing facilities for mammalian cell culture and guidance with the methods, and Hongtao Liu for plasmid DNA.

## Author contributions

W.L. and G.I.J. designed the research project. W.L. performed most of the experiments with contributions from G.G., D.L., C.V. and J.P.. A.H. assisted W.L. in production of transgenic lines. G.G., A.H. and D.L. contributed equally and are listed alphabetically. D.J.L. performed mass spectrometry analysis and S.N. performed HPLC analysis. U.H. provided mutant and transgenic seeds. G.I.J. and J.M.C. obtained funding for the research and managed the project. W.L. and G.I.J. prepared the manuscript, with input from other authors.

## Competing interests

The authors declare no competing interests.
