## [Peer Review File · Nature Communications]

Phosphorylation of Arabidopsis UVR8 photoreceptor
modulates protein interactions and responses to UV-B
radiationReviewer #1 (Remarks to the Author):

Liu et al reported the phosphorylation of UV-B receptor UVR8 protein and its impact on UV-B responses in Arabidopsis in this manuscript. The authors reported that UVR8 is phosphorylated in white light and UV-B supplementation further enhances its phosphorylation status. They identified several putative phosphorylation sites in UVR8 via Mass spectrometry and focused on a well conserved one S402 for further analyses. The authors found that S402 is a major phosphorylation site and well conserved in different plant species. Through analyses the transgenic lines expressing either S402A or S402D, they found that phosphorylation of S402 did not alter the binding of photoreceptor to COP1 and WRKY36, but enhanced the binding to two major negative regulators RUP1 and RUP2 in plants. Interestingly, the phosphorylation of S402 is enhanced in nucleus. Moreover, SPA proteins are required for the phosphorylation of S402 with unknown mechanisms. Finally, the authors found that plant lines expressing S402D accumulate higher levels of HY5 and CHS protein, as well as several secondary metabolites, in response to UV-B.

This is the first report of the phosphorylation of UV-B receptor and its influence on UV-B responses. This work provided significant progress in the understanding of post-translational modification of plant photoreceptor UVR8, which is not much explored. The experiments were well designed and the manuscript is well presented.

There are several issues I raised below for the authors to consider.

1. The authors state that S402 phosphorylation modifies protein interactions with UVR8. However, there are only one line of evidence to support this conclusion (Refer to Figure 3). It would be necessary to test PPI with different approaches to demonstrate the altered interactions (with RUP1 and RUP2) and the unchanged interaction (with COP1).
2. Fig 2e: SPA proteins are required for S402 phosphorylation. Whether the kinase activity of SPAs is important for S402 phosphorylation? Moreover, SPAs has no obvious effect on S402 phosphorylation in the absence of UV-B?
3. It is well established that RUP1 and RUP2 interact with UVR8 and promotes UVR8 rebinding to form homodimer. S402D has increased binding to RUP1 and RUP2. It is very surprising that S402A and S402D have similar photocycle dynamics as the wildtype UVR8.
4. The authors generated multiple S402A and S402D lines as shown in Supple Fig 3. It would be necessary to analyze those lines in UV-B responses, in particular hypocotyl length measurement.
5. Fig 2c: the data here showed that UV-B has no obvious effect on S402 phosphorylation in the NLS-GFP-UVR8 line. Whether UV-B alters nuclear localization of UVR8 in this line?

Reviewer #2 (Remarks to the Author):

The work by Liu et al. entitled "Phosphorylation of Arabidopsis UVR8 photoreceptor modulates protein interactions and responses to UV-B radiation" shows that a small portion of the Arabidopsis UV-B receptor is phosphorylated as a result of UV-B irradiation. This PTM in S402 increases the interaction between UVR8 and its negative regulators RUP1 and RUP2. S402 phosphorylation could also stabilize some enzymes of flavonoid biosynthesis, slightly modifying flavonoid content. This is the first report about a PTM of the master regulator UVR8. This topic is very interesting and suitable to be considered for publication However, there are several points that should be addressed.

I attach below my comments and suggestions, with the intention to help the authors improve their manuscript.

Introduction

The introduction is well-written and clear; the information properly contextualizes this work. The aim of the research is well-defined.

However, in lines 85-88: I believe that results do not support the conclusion that SPA is the kinase involved (my arguments are given below)

Materials and Methods

In general, the methodology is sound and concise, and the work is sufficiently detailed to be reproduced.

My main comment is on the method used to quantify the relative ³²P-labeling of the UVR8 bands (lines 386-395 and Supplementary Table 1). I believe that it will be improved. The intensity of the bands in the autoradiograph was estimated by IMAGE J and "normalised for differences in the amount of radioactivity taken up by the seedlings and also for differences in the amount of immunoprecipitated UVR8 cpm incorporated" (lines 392-394). In addition, some results are the mean of only two experiments (lines 737-740). How do the authors ensure that the autoradiograph exposure is similar in the different experiments to ensure the reproducibility of the calculations? The autoradiograph cpm should be quantified using phosphoimaging technology (e.g., Typhoon FLA 7000) to obtain reliable data. If this is not possible, the results should be reported as qualitative only.

Results

The manuscript is clearly organized, the results are clearly exposed, and the images are of good quality. The supplementary information and raw data were very useful for a complete review of the work.

Nonacclimated plants were exposed to UV-B for 6 to 24 h (Fig. 1 and Supp Table 1).

Mass spectrometry of phosphopeptides was performed on plants exposed for 24 h (Supp table 2)

Overall, these results clearly show that UVR8 is phosphorylated, mainly in S402.

In lines 105 to 107, the authors conclude that the ³²P into UVR8 increased approximately 1.5-fold after 24 hours of UV-B exposure.

I have already outlined why I believe this could be exposed as a qualitative result.

In addition, phosphorylation was analyzed after 6 hours of UV-B exposure. However, the authors show that the UVR8-regulated UV-B response is triggered earlier (see Liao et al, *New Phytologist* (2020) 227: 857–866. doi: 10.1111/nph.16581). It would be illustrative if the authors explained why they chose different irradiation times for the experiments in this manuscript.

Results with the anti-UVR8S402-P antibody confirm the site of UVR8 phosphorylation in Figs. 2a and b (although different UV-B irradiation times were used) and Supp Fig 2a.

From the results in Fig. 2 c and e, the authors conclude that UVR8 is phosphorylated in the nucleus by SPA kinases. After careful analysis of the raw figures in the data set, I have some questions

- Why is the band in the irradiated NLS-GFP -UVR8 more diffuse than in the nonirradiated one?

- The GFP-UVR8-P signal is strongly visible in the autoradiograph (Fig. 1). Did the authors consider using this method instead of the UVR8S402-P antibody to evaluate the results in Fig. 2c and e? In addition, phosphorimaging analysis would significantly improve the assay.

- The UVR8S402-P antibody gives a weak signal in irradiated spa mutants (Fig. 2e), suggesting that this is the kinase that phosphorylates UVR8. To my knowledge, there is no evidence in the literature for a physical interaction between UVR8 and SPA proteins. So how could UVR8 be phosphorylated by SPA? Moreover, the localization of NLS-GFP -UVR8 is an artificial procedure. The authors state in lines 152-154: "The failure of the latter antibodies to detect the upper UVR8S402-P band indicates that UVR8S402-P is a small fraction of total UVR8; in routine western blots with anti-UVR8 antibodies, UVR8S402-P is undetectable."

So how can they be sure that UVR8-P is nuclear? Is it possible that UVR8-P remains cytosolic and is undetectable by UVR8 antibodies? Is it possible to do an autoradiograph or protein gel blot of cytosolic and nuclear fractions to locate UVR8-P?

The generation of the phospho-mimic mutant GFP-UVR8S402D was useful in identifying the importance of phosphorylation in UVR8 protein interactions. In this section of the work, the authors show that UVR8-P interacts mainly with its negative regulators RUP1 and 2 (Fig. 3 and supp Fig. 4).

In lines 204-206, the authors argue, "Hence, any functional consequence of increased RUP binding caused by S402 phosphorylation may not be explained by an effect on UVR8 dimer/monomer status." Conversely, it is not possible to know whether the phosphorylated form is the monomeric or dimeric form. Although UV-B-irradiated UVR8 is mainly nuclear, it cannot be excluded that the small fraction of phosphorylated UVR8 could be a dimer. This should be considered in the model

(Fig. 5)?

Given the laboratory's vast experience with this type of experiment, I think it will be very useful to see GFP-UVR8 phosphorylation during WL recovery.

Finally, the authors studied the accumulation of phenolic compounds in UV-B-acclimated plants growing under photoperiodic conditions. They show that phosphorylation can stabilize CHS by inhibiting proteolysis and increasing the content of phenolic compounds in tissue extracts.

Discussion

The discussion is clear and concise, yet comprehensive. The results have been analyzed in-depth and placed in sufficient context, taking into account previous work. I hope that the entire manuscript (including the discussion) can be rewritten and greatly improved if the authors consider and respond to the comments in this review.

I believe that the model proposed in Fig. 5 needs more experimental support. This could be made clear in the text. For example, there is no clear evidence for phosphorylation of monomers or dimers. Nor whether UVR8-P is mainly nuclear or cytosolic. In addition, it would be very interesting if the authors could speculate on the dephosphorylation of UVR8-P. Does it occur before or simultaneously with dimerization?

This is the first report of a PTM in the master regulator UVR8. This is an important advance in understanding the molecular response of plants to UV-B.

The GFP-UVR8S402D and GFP-UVR8S402A lines generated in this research will be used to gain insights into the importance of UVR8 phosphorylation in physiological responses to UV-B. For example, photomorphogenesis will be easily evaluated, as well as flowering time, UV-B adaptation, the response of acclimated lines, etc. could be also analyzed in future work.

Reviewer #3 (Remarks to the Author):

In this study, the authors investigate a UVR8 mutant in which S402 is mutated to an Ala residue, rendering it unable to be phosphorylated. Initially, the authors demonstrate that UVR8 incorporates about 1.5 times more ³²P within 24 hours of UV-B exposure. They propose that phosphorylation of Ser 402 enhances the binding of RUP proteins to UVR8, thereby modulating the UV-B response. However, it is important to note that Ser 402 phosphorylation is not regulated by UV-B, as this residue is constitutively phosphorylated. Furthermore, the expression of the S402A mutant of UVR8 does not exhibit significant differences compared to the line expressing the equivalent WT construct, including interactions with RUP proteins. Here are my major concerns and suggestions regarding the manuscript:

1. In the abstract, the authors state, "Notably, phosphorylation of Serine 402 strongly promotes binding of REPRESSOR OF UV-B PHOTOMORPHOGENESIS (RUP) proteins, which negatively regulate UVR8 action." However, in the results section (lines 115-117), the authors indicate that Ser 402 is constitutively phosphorylated regardless of the presence or absence of UV-B (data provided in Supplementary Table 2). Therefore, it is unclear how this study addresses the differential interaction of UVR8 with other effectors, as Ser 402 phosphorylation is not UV-B regulated but the UVR8-RUP interaction is regulated by UV-B both in the WT and S402A mutant. Additionally, the authors conclude the abstract with the following statement: "This research provides a basis to understand how UVR8 interacts differentially with numerous effector proteins to regulate diverse responses to UV-B radiation in natural populations and crop species." I do not understand on which experimental basis the authors can reach this conclusion.

2. The authors present data suggesting that they have a Ser402 phospho-specific antibody. However, the data presented in Figure 2 indicates that the antibody recognizes both the

phosphorylated and unphosphorylated forms of UVR8. After phosphatase treatment, the faster migrating isoform is still efficiently recognized by the antibody (Figure 2b). The authors should clarify that this antibody can detect both phosphorylated and unphosphorylated UVR8. The data in Figure 2a, comparing WT and S402A mutant, is consistent with the antibody better recognizing WT UVR8. However, this does not necessarily imply that phosphorylation of Ser 402 is responsible for the observed difference.

3. Related to the previous issue, the blot shown in Figure 2A indicates that GFP-UVR8 migrates as two isoforms both with and without UV-B treatment. Consequently, it is difficult to attribute the mobility shift observed in some gels/blots to UV-B-induced phosphorylation. This point needs clarification in the paper, particularly considering that Ser 402 phosphorylation is not regulated by UV-B. Furthermore, the S402A mutant shown on the same blot is only present as the faster migrating isoform, suggesting that the slower migrating form depends on Ser 402 but not on the presence of UV-B. This observation supports the notion that Ser 402 phosphorylation is not a UV-B regulated process, which is consistent with point 1.

4. Following the arguments presented, if the authors do not identify UV-B regulated phosphorylation sites and study the consequences of such modifications, it is unclear how this study pertains to the light-regulation of UVR8 function. Therefore, the authors need to substantially revise their manuscript, including the abstract, to clarify that they are studying the effects of a mutant S402A that eliminates a constitutive phosphorylation site.

5. According to the model proposed by the authors (and stated in the abstract), phosphorylation of Ser 402 enhances the interaction between UVR8 and RUP proteins. However, the data presented in Figure 3 does not show an obvious difference in the UV-B-induced RUP-UVR8 interaction between the WT and the S402A mutant (Figure 3a). The S402D mutant exhibits stronger interaction than the WT, and this interaction is UV-B dependent (hence unrelated to Ser 402 phosphorylation). Interpreting the role of phospho-mimic mutants, such as S402D, is more challenging than studying a conservative substitution like mutating Ser to Ala. Therefore, the S402D does not provide compelling evidence for the involvement of Ser 402 phosphorylation in regulating UVR8-RUP interaction. This needs to be clarified in the manuscript.

6. The *in vivo* characterization of the S402A and S402D lines does not reveal any significant differences between the WT and S402A line (Figure 4). Although the authors repeatedly highlight significant differences between the S402A and S402D lines, the comparison between WT and S402A is not thoroughly addressed. Small phenotypic differences are observed in the line expressing S402D, but as mentioned earlier, it is difficult to conclude that this pertains specifically to Ser 402 phosphorylation. Furthermore, based on my understanding, the authors report phenotypes using a single WT, S402A, and S402D line. To ensure that differences can be attributed to the transgene, it is necessary to characterize the phenotype of at least two independent lines.

7. Regarding the phenotypic characterization of these lines, the authors' data suggests that the S402D lines have more HY5 proteins. However, this does not lead to higher CHS transcript levels, which is a known HY5 target in UV-B treated plants. The authors then demonstrate that MG132 treatment leads to increased CHS expression. However, it is challenging to evaluate how this observation pertains to the UV-B response and how it relates to the data from their transgenic lines. The authors should provide a clearer explanation regarding the implications of their observations on the UV-B response and their relevance to the data from the transgenic lines.

8. Figure 2e shows samples probed with two different antibodies. The panel appears to have been assembled by combining two halves (IP part), as there is more space and less background in the middle. If this is indeed the case, it must be clearly indicated. Furthermore, if the two halves are from different blots, the authors should refrain from making arguments about different migrations (red dotted line) since this requires side-by-side samples on the same gel/blot.

9. It is worth noting that the best-understood process by which UVR8 regulates transcriptional reprogramming is through its interaction with COP1. The study by Lau et al., 2019 (DOI: 10.15252/embj.2019102140), provides atomic resolution insights into how UVR8 binds to the substrate binding motif of COP1, preventing COP1 from interacting with its substrates (e.g., HY5). It is surprising that this study is not cited in the current manuscript, especially since Ser 402 is in close proximity to the UVR8 peptide that interacts with COP1 (starting with R406). The authors should discuss this study, as it is relevant to their research since they find that COP1 interacts similarly in a UV-B regulated manner with WT, S402A, and S402D mutants.

We thank the reviewers for giving their time to provide insightful comments on our manuscript which have helped us to improve the paper. We have undertaken the additional experimentation suggested and include 8 new figure panels/supplementary figures in the revised submission. Our responses to the reviewers' comments are below:

Reviewer #1 (Remarks to the Author):

Liu et al reported the phosphorylation of UV-B receptor UVR8 protein and its impact on UV-B responses in Arabidopsis in this manuscript. The authors reported that UVR8 is phosphorylated in white light and UV-B supplementation further enhances its phosphorylation status. They identified several putative phosphorylation sites in UVR8 via Mass spectrometry and focused on a well conserved one S402 for further analyses. The authors found that S402 is a major phosphorylation site and well conserved in different plant species. Through analyses the transgenic lines expressing either S402A or S402D, they found that phosphorylation of S402 did not alter the binding of photoreceptor to COP1 and WRKY36, but enhanced the binding to two major negative regulators RUP1 and RUP2 in plants. Interestingly, the phosphorylation of S402 is enhanced in nucleus. Moreover, SPA proteins are required for the phosphorylation of S402 with unknown mechanisms. Finally, the authors found that plant lines expressing S402D accumulate higher levels of HY5 and CHS protein, as well as several secondary metabolites, in response to UV-B.

This is the first report of the phosphorylation of UV-B receptor and its influence on UV-B responses. This work provided significant progress in the understanding of post-translational modification of plant photoreceptor UVR8, which is not much explored. The experiments were well designed and the manuscript is well presented.

Response: We thank the reviewer for the positive comments.

There are several issues I raised below for the authors to consider.

1. The authors state that S402 phosphorylation modifies protein interactions with UVR8. However, there are only one line of evidence to support this conclusion (Refer to Figure 3). It would be necessary to test PPI with different approaches to demonstrate the altered interactions (with RUP1 and RUP2) and the unchanged interaction (with COP1).

Response: We previously focused on interaction in plants using stable Arabidopsis transgenic lines (Fig. 3) and confirmed the increased interaction of RUP2 with UVR8-S402D in transient expression in Nicotiana (Supplementary Fig. 5). However, we accept the reviewer's point and have assayed interactions of proteins expressed in a non-plant system, namely mammalian cells. The results of these experiments support the observations in the original manuscript: the interaction with RUP proteins, notably RUP2, with S402D is enhanced relative to S402A and wild-type UVR8, whereas the UV-B-stimulated interaction between UVR8 and COP1 is unchanged. These data are presented in a new Supplementary Fig. 6.

2. Fig 2e: SPA proteins are required for S402 phosphorylation. Whether the kinase activity of SPAs is important for S402 phosphorylation? Moreover, SPAs has no obvious effect on S402 phosphorylation in the absence of UV-B?

Response: To test whether SPA kinase activity is required for S402 phosphorylation we have used mutants lacking particular SPA proteins and the kinase-dead SPA1-R517E mutant expressed in the *spaQn* background (ms ref. 39). The experiment, presented as a new Fig. 2g and Supplementary Fig. 2d, shows that SPA1 kinase activity is required for maximal, UV-B stimulated S402 phosphorylation. Residual S402 phosphorylation is seen in the absence of UV-B in the *spaQ* mutant (Fig. 2g), suggesting that another kinase may be involved in

S402 phosphorylation. Since SPA is nuclear localised, we propose that S402 is phosphorylated at a low level in the cytosol in the absence of UV-B and that SPA-dependent nuclear S402 phosphorylation occurs following UV-B-stimulated nuclear accumulation.

3. *It is well established that RUP1 and RUP2 interact with UVR8 and promotes UVR8 rebinding to form homodimer. S402D has increased binding to RUP1 and RUP2. It is very surprising that S402A and S402D have similar photocycle dynamics as the wildtype UVR8.*

Response: We agree that altered kinetics of UVR8 re-dimerization might be expected in the mutants based on the existing model of RUP involvement. However, we have assayed re-dimerisation in several experiments and cannot detect any difference between wild-type GFP-UVR8, S402A and S402D lines. This finding suggests that the present model may be too simple to fully explain the re-dimerisation process; it seems that enhanced RUP interaction, on its own, is insufficient to accelerate the kinetics. It is clear that we do not fully understand the roles of RUP proteins.

4. *The authors generated multiple S402A and S402D lines as shown in Supple Fig 3. It would be necessary to analyze those lines in UV-B responses, in particular hypocotyl length measurement.*

Response: We have used at least 2 different transgenic lines for the experiments presented in the paper and have re-worded the Methods to make this clear. We have added a new Supplementary Fig. 4 showing key experiments (protein interactions; CHS protein accumulation) with different transgenic lines, demonstrating that the lines produce equivalent results. We examined hypocotyl extension, but did not find any clear difference between S402A and S402D in several experiments (e.g. Figure 1 below). Hence, we did not include the data in the original manuscript and have not included these data in the revised version because we do not think they enhance the paper; however, we would do so if the reviewers think we should. Our present observations indicate that not all UVR8-mediated responses are altered in S402A and S402D lines. This may be because S402 phosphorylation differentially affects interactions with transcription factors involved in different responses, as noted in the Discussion. We are presently examining which UVR8-mediated responses are affected by S402 phosphorylation and intend to publish the results in subsequent papers – Reviewer 2 notes that the present paper provides a basis for future work. Nevertheless, we have added a new Figure panel (Fig. 4f) showing an additional response modified by S402 phosphorylation, which is an increase in *CHS* transcript accumulation following transfer to elevated UV-B.

Figure 1. Hypocotyl lengths of UVR8 lines. Plants were grown from seed in 0.5 $\mu\text{mol m}^{-2} \text{s}^{-1}$ white light with supplementary 0.25 $\mu\text{mol m}^{-2} \text{s}^{-1}$ UV-B.

5. *Fig 2c: the data here showed that UV-B has no obvious effect on S402 phosphorylation in the NLS-GFP-UVR8 line. Whether UV-B alters nuclear localization of UVR8 in this line?*

Response: As reported previously (ms ref. 12), the NLS-GFP-UVR8 line shows totally constitutive nuclear localisation irrespective of the presence/absence of UV-B.

Reviewer #2 (Remarks to the Author):

The work by Liu et al. entitled "Phosphorylation of Arabidopsis UVR8 photoreceptor modulates protein interactions and responses to UV-B radiation" shows that a small portion of the Arabidopsis UV-B receptor is phosphorylated as a result of UV-B irradiation. This PTM in S402 increases the interaction between UVR8 and its negative regulators RUP1 and RUP2. S402 phosphorylation could also stabilize some enzymes of flavonoid biosynthesis, slightly modifying flavonoid content.

This is the first report about a PTM of the master regulator UVR8. This topic is very interesting and suitable to be considered for publication. However, there are several points that should be addressed.

I attach below my comments and suggestions, with the intention to help the authors improve their manuscript.

Response: We are grateful for the reviewer's constructive comments.

Introduction

The introduction is well-written and clear; the information properly contextualizes this work. The aim of the research is well-defined.

However, in lines 85-88: I believe that results do not support the conclusion that SPA is the kinase involved (my arguments are given below)

Response: In the original manuscript we concluded that SPA protein(s) is/are *required* for S402 phosphorylation and not that they are the kinase that directly phosphorylates UVR8 – although we raised this as a possibility. We now show that SPA1 kinase activity is required (see above); however, although it could be the kinase that phosphorylates UVR8-S402 in the nucleus, it remains possible that it indirectly leads to S402 phosphorylation through direct phosphorylation of another, unidentified substrate, and we state this in the revised manuscript.

Materials and Methods

In general, the methodology is sound and concise, and the work is sufficiently detailed to be reproduced.

My main comment is on the method used to quantify the relative ³²P-labeling of the UVR8 bands (lines 386-395 and Supplementary Table 1). I believe that it will be improved. The intensity of the bands in the autoradiograph was estimated by IMAGE J and "normalised for differences in the amount of radioactivity taken up by the seedlings and also for differences in the amount of immunoprecipitated UVR8 cpm incorporated" (lines 392-394). In addition, some results are the mean of only two experiments (lines 737-740). How do the authors ensure that the autoradiograph exposure is similar in the different experiments to ensure the reproducibility of the calculations? The autoradiograph cpm should be quantified using phosphoimaging technology (e.g., Typhoon FLA 7000) to obtain reliable data. If this is not possible, the results should be reported as qualitative only.

Response: We do not have access to the Typhoon instrument mentioned. Autoradiography was undertaken using the standard method of film exposure, which has been used in numerous previous publications (e.g. Qu *et al.*, 2021, *Plant Physiol.* 187, 917-930). In our study, the same amount of radioactivity was used with the same number of seedlings and the same uptake period in replicate experiments, and the film was exposed for the same time to develop the autoradiographs. Scintillation counting was used to measure radioactivity taken up by the seedlings and the western blot images were made using a Vilber Fusion FX system. As can be seen from the error bars in Fig. 1b, variability between experiments was relatively low. We are confident with the conclusions regarding the stimulation by UV-B exposure and the difference in labelling of mutants. Nevertheless, we accept the reviewer's

point that the data obtained are not absolutely quantitative, and have used the word 'approximately' when referring to the values reported in the text.

Results

The manuscript is clearly organized, the results are clearly exposed, and the images are of good quality. The supplementary information and raw data were very useful for a complete review of the work.

Nonacclimated plants were exposed to UV-B for 6 to 24 h (Fig. 1 and Supp Table 1). Mass spectrometry of phosphopeptides was performed on plants exposed for 24 h (Supp table 2)

Overall, these results clearly show that UVR8 is phosphorylated, mainly in S402.

In lines 105 to 107, the authors conclude that the ^{32}P into UVR8 increased approximately 1.5-fold after 24 hours of UV-B exposure.

I have already outlined why I believe this could be exposed as a qualitative result.

Response: We did not wish to place too much emphasis on the quantification, which is why we stated 'approximately 1.5-fold'.

In addition, phosphorylation was analyzed after 6 hours of UV-B exposure. However, the authors show that the UVR8-regulated UV-B response is triggered earlier (see Liao et al, New Phytologist (2020) 227: 857–866. doi: 10.1111/nph.16581). It would be illustrative if the authors explained why they chose different irradiation times for the experiments in this manuscript.

Response: Our initial objective in this study was to examine whether S402 phosphorylation increases following UV-B exposure and to determine the localisation and regulatory role of the process and we therefore selected timepoints where a clear response could be seen. We have shown that S402 phosphorylation occurs within 6 hours of UV-B exposure and is maintained during 24 h exposure (Fig. 2). Different elements of UV-B photoreception and signaling occur at different timescales after initial exposure and it will be valuable to obtain more information about the kinetics of UVR8 phosphorylation in future studies.

Results with the anti-UVR8S402-P antibody confirm the site of UVR8 phosphorylation in Figs. 2a and b (although different UV-B irradiation times were used) and Supp Fig 2a. From the results in Fig. 2 c and e, the authors conclude that UVR8 is phosphorylated in the nucleus by SPA kinases. After careful analysis of the raw figures in the data set, I have some questions

- Why is the band in the irradiated NLS-GFP-UVR8 more diffuse than in the nonirradiated one?

Response: We do not see a difference between the +/- UV-B bands for NLS-GFP-UVR8 in Fig. 2c and have not seen any difference in other experiments. The NLS-GFP-UVR8 bands appear feinter than the GFP-UVR8 bands because the line has a lower level of expression.

- The GFP-UVR8-P signal is strongly visible in the autoradiograph (Fig. 1). Did the authors consider using this method instead of the UVR8S402-P antibody to evaluate the results in Fig. 2c and e? In addition, phosphorimaging analysis would significantly improve the assay.

Response: The *in vivo* labelling experiments involved quite high levels of radioactivity and we did not want to undertake these experiments more than necessary. Also, the GFP-UVR8 band will be labelled at all phosphorylation sites, and we used the phospho-antibody to focus specifically on S402 phosphorylation. The phospho-antibody gives a clear band for S402-P

and it is also possible to compare the relative intensity of the S402-P and non-phosphorylated UVR8 bands in the same sample.

- The UVR8S402-P antibody gives a weak signal in irradiated spa mutants (Fig. 2e), suggesting that this is the kinase that phosphorylates UVR8. To my knowledge, there is no evidence in the literature for a physical interaction between UVR8 and SPA proteins. So how could UVR8 be phosphorylated by SPA?

Response: As noted above, it is not clear whether SPA directly phosphorylates UVR8, although it is possible that it could. SPA proteins co-immunoprecipitate with UVR8 in Arabidopsis (as do COP1 and RUP proteins), indicating that UVR8 and SPA proteins are members of a protein complex (Heijde et al., 2013, *PNAS* 110, 20326-). Thus, SPA could be sufficiently closely associated with UVR8 in the plant nucleus for direct phosphorylation to occur even though it doesn't show a direct interaction in yeast 2-hybrid experiments.

Moreover, the localization of NLS-GFP-UVR8 is an artificial procedure. The authors state in lines 152-154: "The failure of the latter antibodies to detect the upper UVR8S402-P band indicates that UVR8S402-P is a small fraction of total UVR8; in routine western blots with anti-UVR8 antibodies, UVR8S402-P is undetectable."

So how can they be sure that UVR8-P is nuclear? Is it possible that UVR8-P remains cytosolic and is undetectable by UVR8 antibodies? Is it possible to do an autoradiograph or protein gel blot of cytosolic and nuclear fractions to locate UVR8-P?

Response: We have further addressed the question of localisation of UVR8-S402-P. As suggested by the reviewer, we isolated cytosolic and nuclear fractions and probed with the anti-S402-P antibody. The results, shown in new Fig. 2e show that nuclear GFP-UVR8 is strongly phosphorylated at S402. We think these data are an important addition to the paper and thank the reviewer for the suggestion.

The generation of the phospho-mimic mutant GFP-UVR8S402D was useful in identifying the importance of phosphorylation in UVR8 protein interactions. In this section of the work, the authors show that UVR8-P interacts mainly with its negative regulators RUP1 and 2 (Fig. 3 and supp Fig. 4).

In lines 204-206, the authors argue, "Hence, any functional consequence of increased RUP binding caused by S402 phosphorylation may not be explained by an effect on UVR8 dimer/monomer status". Conversely, it is not possible to know whether the phosphorylated form is the monomeric or dimeric form. Although UV-B-irradiated UVR8 is mainly nuclear, it cannot be excluded that the small fraction of phosphorylated UVR8 could be a dimer. This should be considered in the model (Fig. 5)?

Given the laboratory's vast experience with this type of experiment, I think it will be very useful to see GFP-UVR8 phosphorylation during WL recovery.

Response: We have included a new Supplementary Fig. 2c, which shows that both UVR8 dimer and monomer can be phosphorylated. As suggested by the reviewer, we have modified the model (Figure 5) to recognise this. Also, as requested by the reviewer, we have included a new Figure panel (Fig. 2c) showing S402 phosphorylation status following transfer from UV-B to white light. The experiment shows that de-phosphorylation is incomplete after 6 hours, and is therefore slower than re-dimerisation, which is normally complete within 1-2 hours *in vivo*. However, a comparison of the kinetics is complicated because we do not know whether de-phosphorylation occurs principally in the nucleus or cytosol, and the kinetics of re-dimerisation are measured for total (mainly cytosolic) UVR8. We agree with the reviewer that there are interesting questions to answer here, but we think they should be addressed in future studies.

Finally, the authors studied the accumulation of phenolic compounds in UV-B-acclimated

plants growing under photoperiodic conditions. They show that phosphorylation can stabilize CHS by inhibiting proteolysis and increasing the content of phenolic compounds in tissue extracts.

Discussion

The discussion is clear and concise, yet comprehensive. The results have been analyzed in depth and placed in sufficient context, taking into account previous work. I hope that the entire manuscript (including the discussion) can be rewritten and greatly improved if the authors consider and respond to the comments in this review.

I believe that the model proposed in Fig. 5 needs more experimental support. This could be made clear in the text. For example, there is no clear evidence for phosphorylation of monomers or dimers. Nor whether UVR8-P is mainly nuclear or cytosolic. In addition, it would be very interesting if the authors could speculate on the dephosphorylation of UVR8-P. Does it occur before or simultaneously with dimerization?

Response: As noted above, we have addressed these points in the revised manuscript, and have revised Figure 5.

This is the first report of a PTM in the master regulator UVR8. This is an important advance in understanding the molecular response of plants to UV-B.

The GFP-UVR8S402D and GFP-UVR8S402A lines generated in this research will be used to gain insights into the importance of UVR8 phosphorylation in physiological responses to UV-B. For example, photomorphogenesis will be easily evaluated, as well as flowering time, UV-B adaptation, the response of acclimated lines, etc. could be also analyzed in future work.

Reviewer #3 (Remarks to the Author):

In this study, the authors investigate a UVR8 mutant in which S402 is mutated to an Ala residue, rendering it unable to be phosphorylated. Initially, the authors demonstrate that UVR8 incorporates about 1.5 times more ³²P within 24 hours of UV-B exposure. They propose that phosphorylation of Ser 402 enhances the binding of RUP proteins to UVR8, thereby modulating the UV-B response. However, it is important to note that Ser 402 phosphorylation is not regulated by UV-B, as this residue is constitutively phosphorylated.

Response: As shown in the original manuscript and again in the revised manuscript (see Figures 2a, 2b, 2c, 2g) S402 phosphorylation is increased following UV-B exposure. An important point is that the abundance of S402-P in the nucleus, where UVR8 is functional, is strongly enhanced following UV-B exposure (Fig. 2e).

Furthermore, the expression of the S402A mutant of UVR8 does not exhibit significant differences compared to the line expressing the equivalent WT construct, including interactions with RUP proteins.

Response: We address this point in 5. and 6. below.

Here are my major concerns and suggestions regarding the manuscript:

1. In the abstract, the authors state, "Notably, phosphorylation of Serine 402 strongly promotes binding of REPRESSOR OF UV-B PHOTOMORPHOGENESIS (RUP) proteins, which negatively regulate UVR8 action." However, in the results section (lines 115-117), the authors indicate that Ser 402 is constitutively phosphorylated regardless of the presence or absence of UV-B (data provided in Supplementary Table 2). Therefore, it is unclear how this study addresses the differential interaction of UVR8 with other effectors, as Ser 402 phosphorylation is not UV-B regulated but the UVR8-RUP interaction is regulated by UV-B both in the WT and S402A mutant.

Response: The mass spectrometry data (Supplementary Table 2) show that S402 is *detectable* in minus- and plus-UV-B samples. As noted above, experiments using the phospho-antibody (Fig. 2) clearly show that the abundance of UVR8-S402-P is stimulated by UV-B exposure.

Additionally, the authors conclude the abstract with the following statement: "This research provides a basis to understand how UVR8 interacts differentially with numerous effector proteins to regulate diverse responses to UV-B radiation in natural populations and crop species." I do not understand on which experimental basis the authors can reach this conclusion.

Response: In this sentence we draw attention to the broad significance of the research. Our finding that UVR8 is phosphorylated at multiple residues does provide a basis to understand how interactions with other proteins may be regulated, which will have consequences for responses in diverse species.

2. The authors present data suggesting that they have a Ser402 phospho-specific antibody. However, the data presented in Figure 2 indicates that the antibody recognizes both the phosphorylated and unphosphorylated forms of UVR8. After phosphatase treatment, the faster migrating isoform is still efficiently recognized by the antibody (Figure 2b). The authors should clarify that this antibody can detect both phosphorylated and unphosphorylated UVR8. The data in Figure 2a, comparing WT and S402A mutant, is consistent with the antibody better recognizing WT UVR8. However, this does not necessarily imply that phosphorylation of Ser 402 is responsible for the observed difference.

Response: The reviewer is correct that the phospho-antibody recognizes both the phosphorylated and unphosphorylated forms of UVR8, as we originally stated in the paper. The data presented using the S402A mutant (Fig. 2a) and phosphatase treatment (Fig. 2b) demonstrate very clearly that the upper band is S402-phosphorylated UVR8. In contrast to the reviewer's statement, the experiment in Fig. 2a shows that the phospho-antibody more strongly recognises the upper UVR8-S402-P band than the lower unphosphorylated UVR8 band.

3. Related to the previous issue, the blot shown in Figure 2A indicates that GFP-UVR8 migrates as two isoforms both with and without UV-B treatment. Consequently, it is difficult to attribute the mobility shift observed in some gels/blots to UV-B-induced phosphorylation. This point needs clarification in the paper, particularly considering that Ser 402 phosphorylation is not regulated by UV-B. Furthermore, the S402A mutant shown on the same blot is only present as the faster migrating isoform, suggesting that the slower migrating form depends on Ser 402 but not on the presence of UV-B. This observation supports the notion that Ser 402 phosphorylation is not a UV-B regulated process, which is consistent with point 1.

Response: As explained in response to points 1 and 2 above, the data clearly show that the upper band recognised by the phospho-antibody is S402-phosphorylated UVR8 (Figs. 2a and 2b) and its abundance is increased by UV-B exposure (Figures 2a, 2b, 2c, 2e, 2g). The mobility shift observed in the *in vivo* labelling experiments (Figure 1) is consistent with the upper UVR8-S402-P band recognised by the antibody.

4. Following the arguments presented, if the authors do not identify UV-B regulated phosphorylation sites and study the consequences of such modifications, it is unclear how this study pertains to the light-regulation of UVR8 function. Therefore, the authors need to substantially revise their manuscript, including the abstract, to clarify that they are studying the effects of a mutant S402A that eliminates a constitutive phosphorylation site.

Response: Please see the responses above; S402 phosphorylation is stimulated by UV-B, notably in the nucleus where UVR8 is functional.

5. According to the model proposed by the authors (and stated in the abstract), phosphorylation of Ser 402 enhances the interaction between UVR8 and RUP proteins. However, the data presented in Figure 3 does not show an obvious difference in the UV-B-induced RUP-UVR8 interaction between the WT and the S402A mutant (Figure 3a).

Response: The effects of specific phosphosite mutations need to be considered in the broader context of protein structure. Our finding that phospho-null and phosphomimetic mutations have different impacts is not unusual. There are various published examples where phospho-null mutations have little/no functional consequence in contrast to phosphomimetic mutations at the same site (e.g. Satyanarayana et al., 2009, *Drug Metabolism and Disposition* 37, 719-730; Fodor-Dunai et al., 2011, *Plant J.* 66, 669-679; Bush et al., 2016, *Plant Physiol.* 172, 128-140; Liu et al., 2020, *Plant Physiol.* 183, 194-205; Wang et al., 2020, *Int. J. Mol. Scis.* 21, 9183). However, introducing a phosphomimetic mutation will often have a strong effect because of the change in charge. In the specific case of UVR8, although the C27 region is important for protein interactions, it is known that COP1 and RUP proteins also bind to the β -propeller core (ms refs. 16, 21-23). Therefore, the C27 S402A mutation may reduce the strength of protein binding to UVR8 but not prevent it, since some degree of interaction may be retained via the β -propeller region. However, the charge added by the S402D mutation (demonstrated by the mobility shift in the protein - see Fig. 3a Input) has the potential to strengthen interactions, which we see clearly with RUP proteins. Thus, S402A will not necessarily cause a strong interaction or response phenotype because of the residual binding to the core domain, in contrast to S402D.

The S402D mutant exhibits stronger interaction than the WT, and this interaction is UV-B dependent (hence unrelated to Ser 402 phosphorylation).

Response: Regarding the statement in brackets, we have already addressed the point that S402 phosphorylation is stimulated by UV-B.

Interpreting the role of phospho-mimic mutants, such as S402D, is more challenging than studying a conservative substitution like mutating Ser to Ala. Therefore, the S402D does not provide compelling evidence for the involvement of Ser 402 phosphorylation in regulating UVR8-RUP interaction. This needs to be clarified in the manuscript.

Response: The approach of making a phosphomimetic mutant has been used extensively in research to study the effects of protein phosphorylation and is widely regarded as a reliable approach to examine the role of a specific phosphorylated residue. All mutation experiments need to be interpreted with caution, but the results with the S402D lines provide strong evidence that S402 phosphorylation does modify interaction of UVR8 with RUP proteins.

6. The in vivo characterization of the S402A and S402D lines does not reveal any significant differences between the WT and S402A line (Figure 4). Although the authors repeatedly highlight significant differences between the S402A and S402D lines, the comparison between WT and S402A is not thoroughly addressed. Small phenotypic differences are observed in the line expressing S402D, but as mentioned earlier, it is difficult to conclude that this pertains specifically to Ser 402 phosphorylation.

Response: The phenotypic difference between S402A and wild-type will be affected by several factors. Firstly, as mentioned in 5. above, residual protein binding to S402A may mean that the level of response is not greatly different to wild-type. Secondly, the level of S402 phosphorylation in wild-type will determine the extent to which its response resembles

that of S402A or S402D. As explained in the paper, UVR8-S402-P is a small fraction of total cellular UVR8. Hence, the level of S402 phosphorylation in wild-type is likely to be closer to S402A than S402D. Therefore we may not expect the phenotypic difference between S402A and wild-type to be as great as between S402D and wild-type. Moreover, the extent of S402 phosphorylation in wild-type will vary with growth and illumination conditions. This is why we have focused on the difference between S402A and S402D in characterising the response phenotypes. The effects we report on HY5 and CHS protein accumulation are clear, statistically significant and reveal a novel aspect of regulation. Moreover, we show that S402D has an approximately 20-25% increase in flavonoid/HCA content over S402A, which is quite substantial. In addition, the new data in Fig. 4f show a 50% greater fold-increase in *CHS* transcript accumulation in S402D compared to S402A when plants are transferred to elevated UV-B. Together these data provide strong evidence for the physiological relevance of S402 phosphorylation.

Furthermore, based on my understanding, the authors report phenotypes using a single WT, S402A, and S402D line. To ensure that differences can be attributed to the transgene, it is necessary to characterize the phenotype of at least two independent lines.

Response: We stated in the Methods that at least 2 independent lines were used for each genotype. As mentioned in response to Reviewer 1, we have re-worded the Methods to make this clearer and have added a new Supplementary Fig. 4 showing that different transgenic lines produce equivalent results.

7. Regarding the phenotypic characterization of these lines, the authors' data suggests that the S402D lines have more HY5 proteins. However, this does not lead to higher CHS transcript levels, which is a known HY5 target in UV-B treated plants. The authors then demonstrate that MG132 treatment leads to increased CHS expression. However, it is challenging to evaluate how this observation pertains to the UV-B response and how it relates to the data from their transgenic lines. The authors should provide a clearer explanation regarding the implications of their observations on the UV-B response and their relevance to the data from the transgenic lines.

Response: The induction of *CHS* transcription is not solely dependent on HY5; if another key transcription factor is not affected by S402 phosphorylation an increase in *CHS* transcript level may not be observed. However, we have included a new Figure panel, Fig. 4f, which shows that S402 phosphorylation does have an effect on *CHS* transcript level when UV-B exposure is elevated. As explained in the paper, the increased accumulation of CHS protein in the S402D transgenic line correlates with the increased abundance of flavonoids in response to UV-B.

8. Figure 2e shows samples probed with two different antibodies. The panel appears to have been assembled by combining two halves (IP part), as there is more space and less background in the middle. If this is indeed the case, it must be clearly indicated. Furthermore, if the two halves are from different blots, the authors should refrain from making arguments about different migrations (red dotted line) since this requires side-by-side samples on the same gel/blot.

Response: The original Fig. 2e presented two halves of the same blot, loaded with identical samples, probed with different antibodies. However, we have now replaced the original Fig. 2e with a new Figure panel extending the original data (Fig. 2g). We agree that comparisons of band mobilities on different blots need to be done with care, but this is made easier by the fact that the lower non-phosphorylated UVR8 band detected by the phospho-antibody aligns with the band detected by the polyclonal anti-UVR8 antibody.

9. It is worth noting that the best-understood process by which UVR8 regulates

transcriptional reprogramming is through its interaction with COP1. The study by Lau et al., 2019 (DOI: 10.15252/embj.2019102140), provides atomic resolution insights into how UVR8 binds to the substrate binding motif of COP1, preventing COP1 from interacting with its substrates (e.g., HY5). It is surprising that this study is not cited in the current manuscript, especially since Ser 402 is in close proximity to the UVR8 peptide that interacts with COP1 (starting with R406). The authors should discuss this study, as it is relevant to their research since they find that COP1 interacts similarly in a UV-B regulated manner with WT, S402A, and S402D mutants.

Response: We have referred to the impressive Lau et al. study in the revised manuscript (ms ref. 21). We didn't do so previously because it is not possible to cite all relevant papers. The importance of the VP motif in COP1 binding was first noted in a previous paper by Yin et al. (ms ref. 16), and a more recent paper by Wang et al. (ms ref. 22) also gives structural information on the interaction between UVR8 and COP1.

Reviewer #1 (Remarks to the Author):

All my previous comments and suggestions are properly addressed by the authors. I would like to recommend the current manuscript for publication.

Reviewer #2 (Remarks to the Author):

I believe that the authors replied mostly to my comments, and the manuscript was improved. This updated version will be suitable for publication. However, I still have a concern regarding the representation of phosphorylation in Fig 5. It appears that the depiction of UVR8 suggests the entire pool is phosphorylated, creating ambiguity about the actual fraction involved. Despite the clarification in the figure legend (lines 964-66) that highlights the phosphorylation of a small fraction of cytosolic UVR8 and its enhanced phosphorylation in the nucleus, this distinction is not as evident in the visual representation. It would be beneficial to enhance the clarity of the figure to accurately reflect that only a small fraction of the nuclear UVR8 is phosphorylated, as specified in the legend.

Reviewer #2 Attachment on the following page.

Reviewer #3 (Remarks to the Author):

Comment: *In this study, the authors investigate a UVR8 mutant in which S402 is mutated to an Ala residue, rendering it unable to be phosphorylated. Initially, the authors demonstrate that UVR8 incorporates about 1.5 times more 32P within 24 hours of UV-B exposure. They propose that phosphorylation of Ser 402 enhances the binding of RUP proteins to UVR8, thereby modulating the UV-B response. However, it is important to note that Ser 402 phosphorylation is not regulated by UV-B, as this residue is constitutively phosphorylated.*

Response: As shown in the original manuscript and again in the revised manuscript (see Figures 2a, 2b, 2c, 2g) S402 phosphorylation is increased following UV-B exposure. An important point is that the abundance of S402-P in the nucleus, where UVR8 is functional, is strongly enhanced following UV-B exposure (Fig. 2e).

My opinion: *I agree with the authors. The clarity provided by the new Figure 2 in the updated manuscript resolves any uncertainties regarding the UV-B-induced phosphorylation of UVR8.*

Comment: *Furthermore, the expression of the S402A mutant of UVR8 does not exhibit significant differences compared to the line expressing the equivalent WT construct, including interactions with RUP proteins.*

Response: We address this point in 5. and 6. below.

Comment: *Here are my major concerns and suggestions regarding the manuscript:*

Comment:1. *In the abstract, the authors state, "Notably, phosphorylation of Serine 402 strongly promotes binding of REPRESSOR OF UV-B PHOTOMORPHOGENESIS (RUP) proteins, which negatively regulate UVR8 action." However, in the results section (lines 115-117), the authors indicate that Ser 402 is constitutively phosphorylated regardless of the presence or absence of UV-B (data provided in Supplementary Table 2). Therefore, it is unclear how this study addresses the differential interaction of UVR8 with other effectors, as Ser 402 phosphorylation is not UV-B regulated but the UVR8-RUP interaction is regulated by UV-B both in the WT and S402A mutant.*

Response: The mass spectrometry data (Supplementary Table 2) show that S402 is detectable in minus- and plus-UV-B samples. As noted above, experiments using the phospho-antibody (Fig. 2) clearly show that the abundance of UVR8-S402-P is stimulated by UV-B exposure.

My opinion: *I agree with the authors. The mass spectrometry data and the new Figure 2 in the updated manuscript resolve any uncertainties regarding the UV-B-induced phosphorylation of UVR8.*

Comment: *Additionally, the authors conclude the abstract with the following statement: "This research provides a basis to understand how UVR8 interacts differentially with numerous effector proteins to regulate diverse responses to UV-B radiation in natural populations and crop species." I do not understand on which experimental basis the authors can reach this conclusion.*

Response: In this sentence, we draw attention to the broad significance of the research. Our finding that UVR8 is phosphorylated at multiple residues does provide a basis to understand how interactions with other proteins may be regulated, which will have consequences for responses in diverse species.

My opinion *I agree with the Reviewer's observation that this statement is speculative. From my perspective, there lacks a convincing physiological counterpart for the UVR8 phosphorylation in this study (see below)*

Comment 2. *The authors present data suggesting that they have a Ser402 phospho-specific antibody. However, the data presented in Figure 2 indicates that the antibody recognizes both the phosphorylated and unphosphorylated forms of UVR8. After phosphatase treatment, the faster migrating isoform is still efficiently recognized by the antibody (Figure 2b). The authors should clarify that this antibody can detect both phosphorylated and unphosphorylated UVR8. The data in Figure 2a, comparing WT and S402A mutant, is consistent with the antibody better recognizing WT UVR8. However, this does not necessarily imply that phosphorylation of Ser 402 is responsible for the observed difference.*

Response: The reviewer is correct that the phospho-antibody recognizes both the phosphorylated and unphosphorylated forms of UVR8, as we originally stated in the paper. The data presented using the S402A mutant (Fig. 2a) and phosphatase treatment (Fig. 2b) demonstrate very clearly that the upper band is S402-phosphorylated UVR8. In contrast to the reviewer's statement, the experiment in Fig. 2a shows that the phospho-antibody more strongly recognises the upper UVR8-S402-P band than the lower unphosphorylated UVR8 band.

My opinion *Considering both the previous and latest findings presented in Figure 2 and the supplementary information, I share the authors' opinion that the phospho-antibody effectively recognizes the phosphorylated band.*

Comment 3. *Related to the previous issue, the blot shown in Figure 2A indicates that GFP-UVR8 migrates as two isoforms both with and without UV-B treatment. Consequently, it is difficult to attribute the mobility shift observed in some gels/blots to UV-B-induced phosphorylation. This point needs clarification in the paper, particularly considering that Ser 402 phosphorylation is not regulated by UV-B. Furthermore, the S402A mutant shown on the same blot is only present as the faster migrating isoform, suggesting that the slower migrating form depends on Ser 402 but not on the presence of UV-B. This observation supports the notion that Ser 402 phosphorylation is not a UV-B-regulated process, which is consistent with point 1.*

Response: As explained in response to points 1 and 2 above, the data clearly show that the upper band recognised by the phospho-antibody is S402-phosphorylated UVR8 (Figs. 2a and 2b) and its abundance is increased by UV-B exposure (Figures 2a, 2b, 2c, 2e, 2g). The mobility shift observed in the *in vivo* labelling experiments (Figure 1) is consistent with the upper UVR8-S402-P band recognised by the antibody.

My opinion *I agree with the authors' viewpoint. Furthermore, I believe this is more effectively illustrated in Supplementary Figure 1, where the disparity in migration between Autoradiography and Western Blot is evident.*

Comment:4. *Following the arguments presented, if the authors do not identify UV-B regulated phosphorylation sites and study the consequences of such modifications, it is unclear how this study pertains to the light-regulation of UVR8 function. Therefore, the authors need to substantially revise their manuscript, including the abstract, to clarify that they are studying the effects of a mutant S402A that eliminates a constitutive phosphorylation site.*

Response: Please see the responses above; S402 phosphorylation is stimulated by UV-B, notably in the nucleus where UVR8 is functional.

My opinion *I agree with the authors' viewpoint.*

Comment:5. According to the model proposed by the authors (and stated in the abstract), phosphorylation of Ser 402 enhances the interaction between UVR8 and RUP proteins. However, the data presented in Figure 3 does not show an obvious difference in the UV-B induced RUP-UVR8 interaction between the WT and the S402A mutant (Figure 3a).

Response: The effects of specific phosphosite mutations need to be considered in the broader context of protein structure. Our finding that phospho-null and phosphomimetic mutations have different impacts is not unusual. There are various published examples where phospho-null mutations have little/no functional consequence in contrast to phosphomimetic mutations at the same site (e.g. Satyanarayana et al., 2009, *Drug Metabolism and Disposition* 37, 719-730; Fodor-Dunai et al., 2011, *Plant J.* 66, 669-679; Bush et al., 2016, *Plant Physiol.* 172, 128-140; Liu et al., 2020, *Plant Physiol.* 183, 194-205; Wang et al., 2020, *Int. J. Mol. Scis.* 21, 9183). However, introducing a phosphomimetic mutation will often have a strong effect because of the change in charge. In the specific case of UVR8, although the C27 region is important for protein interactions, it is known that COP1 and RUP proteins also bind to the b-propeller core (ms refs. 16, 21-23). Therefore, the C27 S402A mutation may reduce the strength of protein binding to UVR8 but not prevent it, since some degree of interaction may be retained via the b-propeller region. However, the charge added by the S402D mutation (demonstrated by the mobility shift in the protein - see Fig. 3a Input) has the potential to strengthen interactions, which we see clearly with RUP proteins. Thus, S402A will not necessarily cause a strong interaction or response phenotype because of the residual binding to the core domain, in contrast to S402D.

My opinion: I agree with the authors' assertion that the phosphomimetic exhibits a stronger interaction with RUP proteins. Nevertheless, in agreement with Reviewer#3, I find their explanation for the absence of differences between S402 and WT lacking. While it is true that other studies demonstrate variations between phospho-null and phosphomimetic mutations, this alone does not sufficiently justify the observed lack of distinctions between S402 and WT.

Comment:

The S402D mutant exhibits stronger interaction than the WT, and this interaction is UV-B dependent (hence unrelated to Ser 402 phosphorylation).

Response: Regarding the statement in brackets, we have already addressed the point that S402 phosphorylation is stimulated by UV-B.

My opinion; Authors show that S402 phosphorylation is stimulated by UV-B., but not that this phosphorylation increases the interaction with RUP. That was increased in the phosphomimetic, but not in WT. Moreover, if S402D is phosphomimetic (and behaves like a "constitutively" phosphorylated), why the interaction with RUP is enhanced by UV-B? If the authors consider that this increase will be provided by phosphorylation of other residues than S402, that will be seen in WT and S402A lines.

Comment: Interpreting the role of phospho-mimic mutants, such as S402D, is more challenging than studying a conservative substitution like mutating Ser to Ala. Therefore, the S402D does not provide compelling evidence for the involvement of Ser 402 phosphorylation in regulating UVR8-RUP interaction. This needs to be clarified in the manuscript.

Response: The approach of making a phosphomimetic mutant has been used extensively in research to study the effects of protein phosphorylation and is widely regarded as a reliable

approach to examine the role of a specific phosphorylated residue. All mutation experiments need to be interpreted with caution, but the results with the S402D lines provide strong evidence that S402 phosphorylation does modify interaction of UVR8 with RUP proteins.

My opinion: *Please, refer to my previous comment regarding this matter.*

Comment:6. *The in vivo characterization of the S402A and S402D lines does not reveal any significant differences between the WT and S402A line (Figure 4). Although the authors repeatedly highlight significant differences between the S402A and S402D lines, the comparison between WT and S402A is not thoroughly addressed. Small phenotypic differences are observed in the line expressing S402D, but as mentioned earlier, it is difficult to conclude that this pertains specifically to Ser 402 phosphorylation.*

Response: The phenotypic difference between S402A and wild-type will be affected by several factors. Firstly, as mentioned in 5. above, residual protein binding to S402A may mean that the level of response is not greatly different to wild-type. Secondly, the level of S402 phosphorylation in wild-type will determine the extent to which its response resembles that of S402A or S402D. As explained in the paper, UVR8-S402-P is a small fraction of total cellular UVR8. Hence, the level of S402 phosphorylation in wild-type is likely to be closer to S402A than S402D. Therefore we may not expect the phenotypic difference between S402A and wild-type to be as great as between S402D and wild-type. Moreover, the extent of S402 phosphorylation in wild-type will vary with growth and illumination conditions. This is why we have focused on the difference between S402A and S402D in characterising the response phenotypes. The effects we report on HY5 and CHS protein accumulation are clear, statistically significant and reveal a novel aspect of regulation. Moreover, we show that S402D has an approximately 20-25% increase in flavonoid/HCA content over S402A, which is quite substantial. In addition, the new data in Fig. 4f show a 50% greater fold-increase in CHS transcript accumulation in S402D compared to S402A when plants are transferred to elevated UV-B. Together these data provide strong evidence for the physiological relevance of S402 phosphorylation.

My opinion *GFP-UVR8S402D lines were generated in the uvr8-1 background. However, it's important to note that in vivo phosphorylation is limited to a small fraction of WT UVR8 lines. Complete phosphorylation of the entire UVR8 pool, as simulated in the phosphomimetic S402D mutation, is not achievable. Consequently, for any significant physiological impact, the entire cellular UVR8 pool would need to be phosphorylated. In alignment with the perspective of Reviewer #3, I believe that selecting S402D might not be the optimal choice for attributing a physiological role to UV-B-induced S402 phosphorylation.*

Comment: *Furthermore, based on my understanding, the authors report phenotypes using a single WT, S402A, and S402D line. To ensure that differences can be attributed to the transgene, it is necessary to characterize the phenotype of at least two independent lines.*

Response: We stated in the Methods that at least 2 independent lines were used for each genotype. As mentioned in response to Reviewer 1, we have re-worded the Methods to make this clearer and have added a new Supplementary Fig. 4 showing that different transgenic lines produce equivalent results.

My opinion: *This comment was satisfactorily addressed by the authors.*

Comment:7. *Regarding the phenotypic characterization of these lines, the authors' data suggests that the S402D lines have more HY5 proteins. However, this does not lead to higher CHS transcript levels, which is a known HY5 target in UV-B-treated plants. The authors then*

demonstrate that MG132 treatment leads to increased CHS expression. However, it is challenging to evaluate how this observation pertains to the UV-B response and how it relates to the data from their transgenic lines. The authors should provide a clearer explanation regarding the implications of their observations on the UV-B response and their relevance to the data from the transgenic lines.

Response: The induction of CHS transcription is not solely dependent on HY5; if another key transcription factor is not affected by S402 phosphorylation an increase in CHS transcript level may not be observed. However, we have included a new Figure panel, Fig. 4f, which shows that S402 phosphorylation does have an effect on CHS transcript level when UV-B exposure is elevated. As explained in the paper, the increased accumulation of CHS protein in the S402D transgenic line correlates with the increased abundance of flavonoids in response to UV-B.

My opinion: *Setting aside my comments regarding the validity of using S402D as a tool, the result is correct.*

Comment:8. *Figure 2e shows samples probed with two different antibodies. The panel appears to have been assembled by combining two halves (IP part), as there is more space and less background in the middle. If this is indeed the case, it must be clearly indicated. Furthermore, if the two halves are from different blots, the authors should refrain from making arguments about different migrations (red dotted line) since this requires side-by-side samples on the same gel/blot.*

Response: The original Fig. 2e presented two halves of the same blot, loaded with identical samples, probed with different antibodies. However, we have now replaced the original Fig. 2e with a new Figure panel extending the original data (Fig. 2g). We agree that comparisons of band mobilities on different blots need to be done with care, but this is made easier by the fact that the lower non-phosphorylated UVR8 band detected by the phospho-antibody aligns with the band detected by the polyclonal anti-UVR8 antibody.

My opinion *After a reexamination of the supplementary material, I aligned the two blots utilizing the 40 kDa markers, and I confirmed that the lower non-phosphorylated UVR8 band detected by the phospho-antibody aligns with the band detected by the polyclonal anti-UVR8 antibody. However, I believe it is crucial to emphasize Supplementary Figure 2 to strengthen the authors' argument.*

Comment:9. *It is worth noting that the best-understood process by which UVR8 regulates transcriptional reprogramming is through its interaction with COP1. The study by Lau et al., 2019 (DOI: 10.15252/emj.2019102140), provides atomic resolution insights into how UVR8 binds to the substrate binding motif of COP1, preventing COP1 from interacting with its substrates (e.g., HY5). It is surprising that this study is not cited in the current manuscript, especially since Ser 402 is in close proximity to the UVR8 peptide that interacts with COP1 (starting with R406). The authors should discuss this study, as it is relevant to their research since they find that COP1 interacts similarly in a UV-B regulated manner with WT, S402A, and S402D mutants.*

Response: We have referred to the impressive Lau et al. study in the revised manuscript (ms ref. 21). We didn't do so previously because it is not possible to cite all relevant papers. The importance of the VP motif in COP1 binding was first noted in a previous paper by Yin et al. (ms ref. 16), and a more recent paper by Wang et al. (ms ref. 22) also gives structural information on the interaction between UVR8 and COP1.

My opinion: *OK, I Have no comment*

NCOMMS-23-20262A-Z
Response to Reviewers' Comments

Reviewer #1

All my previous comments and suggestions are properly addressed by the authors. I would like to recommend the current manuscript for publication.

Response: thanks.

Reviewer #2

I believe that the authors replied mostly to my comments, and the manuscript was improved. This updated version will be suitable for publication. However, I still have a concern regarding the representation of phosphorylation in Fig 5. It appears that the depiction of UVR8 suggests the entire pool is phosphorylated, creating ambiguity about the actual fraction involved. Despite the clarification in the figure legend (lines 964-66) that highlights the phosphorylation of a small fraction of cytosolic UVR8 and its enhanced phosphorylation in the nucleus, this distinction is not as evident in the visual representation. It would be beneficial to enhance the clarity of the figure to accurately reflect that only a small fraction of the nuclear UVR8 is phosphorylated, as specified in the legend.

Response: we accept the Reviewer's point, and have modified Figure 5 to indicate visually that only a fraction of the total UVR8 pool becomes phosphorylated in the nucleus.